



# A Novel Approach to Map the Intensity of Surface Melting on the Antarctica Ice Sheet using SMAP L-Band Microwave Radiometry

Seyedmohammad Mousavi[1], Andreas Colliander[1], Julie Z. Miller[2], John S. Kimball[3]

[1]Jet Propulsion Laboratory, California Institute of Technology, Pasadena, CA 91109, USA
[2]Earth Science and Observation Center, University of Colorado, Boulder, CO 80309, USA
[3]Numerical Terradynamic Simulation Group, University of Montana, Missoula, MT 59812, USA

*Correspondence to*: Seyedmohammad Mousavi (mousavi@jpl.nasa.gov)

**Abstract.** The polar ice sheets have undergone unprecedented melt events in the recent past years, which have consequences for ice sheet mass balance, stability, and sea level. In this study, we employed L-band (1.4 GHz) brightness temperature observations collected by NASA's Soil Moisture Active Passive (SMAP) mission to investigate the extent, duration and intensity of melt events on the Antarctic Ice Sheet from 2015 to 2020. Satellite microwave measurements have long been used
to detect melt events because of their sensitivity to the presence of liquid water in snow and ice. The observed microwave response depends on the sensor measurement frequency. Our hypothesis for this study is that the relatively long wavelength SMAP observations can detect a wider range of surface wetness conditions relative to shorter wavelength microwave observations that attain signal saturation at relatively lower wetness levels and within shallower surface layers. SMAP provides nearly all-weather surface monitoring over all of Antarctica twice daily with morning and evening overpasses at about 40 km
spatial resolution. We applied an empirical threshold algorithm using horizontally and vertically polarized microwave brightness temperature differences to detect surface melt events over Antarctica from 2015 through 2020. The results show that the SMAP empirical algorithm can be used to detect melt extent and duration, and the geophysical model-based algorithm can be used to detect snow wetness, which can be used as an indicator of melt intensity. Analysis of the melt seasons between 2015 and 2020 show that even though the melt extent in 2019-2020 was not as large as during the 2015-2016 melt season, it
was significantly more intense, particularity on the West Antarctic Ice Sheet.

## 1   INTRODUCTION

Temperatures are increasing across large parts of Antarctica as a result of climate warming (Steig, et al., 2009). Monitoring the melt extent and duration over Antarctic ice shelves and coastal areas is important for documenting climate change impacts on ice sheet stability and sea level rise (Liu, et al., 2006; Picard et al., 2007; Golledge, et al., 2015). Shepherd et al. (2018)
found that the Antarctic ice loss between 1992 and 2017 corresponds to an increase in mean sea level rise of 7.6 mm. Slater et al. (2020) showed that the Antarctica has been losing ice mass at an increasing rate in the recent decades, while Frederikse et al. (2020) showed that the relative significance of Antarctica's influence on sea level rise will increase in the future. Intense melting over multiple years can result in catastrophic ice shelf collapse and disintegration (Scambos, et al., 2000). The



Antarctic Peninsula, in particular, has experienced several major ice shelf collapses as a result of a significant annual melt
cycle (Datta, et al., 2019).

Due to their all-weather operational capability and sensitivity to the presence of the liquid water in snow, both satellite
microwave radar and radiometer systems are commonly used to detect melt events over Greenland (Ashcraft & Long, n.d.;
Mote & Anderson, 1995; Mote, et al., 1993; Wismann, 2000). Past studies reporting on melt events over Antarctica commonly
used 19 GHz and 37 GHz frequencies available from several satellite sensors, such as the Scanning Multichannel Microwave
Radiometer (SMMR) on the Nimbus 7 satellite or the Special Sensor Microwave/Imager (SSM/I) and Special Sensor
Microwave Imager Sounder (SSMIS) from the Defense Meteorological Satellite Program (DMSP) satellites, since the
beginning of the satellite era (e.g., (Liu, et al., 2006; Picard et al., 2007; Scambos, et al., 2000; Ridley, 1993; Zwally & S.
Fiegles, 1994; Torinesi, et al., 2003).

L-band (1.4 GHz) radiometer systems may provide more comprehensive information on the polar ice sheets because the larger
characteristic ice penetration and sensing depth at lower microwave frequencies, which can extend up to hundreds of meters
(Jezek et al., 2015),  (Leduc-Leballeur, et al., 2020; Jezek et al., 2018; Macelloni et al., 2019; Miller et al., 2020). In this paper,
we investigated the response of the L-band radiometer on the NASA SMAP (Soil Moisture Active Passive) satellite, launched
in January 2015, to Antarctica melt events (Entekhabi, et al., 2010). The objective of the study was to detect dielectric changes
in the surface composition, such as snow wetness percentage, and relate those changes to melt events. Studies have shown that
the surface of the ice sheet may warm to depths of about 3 m during melt events (Munneke, et al., 2018). Our hypothesis was
that the long wavelength measurements from SMAP are more sensitive to a higher melt intensity and deeper surface
snow/firn/ice layers than the shorter wavelength measurements used by more conventional satellite microwave radiometers.
Our approach was to develop an empirical algorithm to detect melt events and a geophysical model-based algorithm to
determine spatial and temporal variations in surface wetness over the Antarctica ice sheet using SMAP brightness temperature
retrievals. This paper is organized as follows. In Sect. 2, we briefly talk about the SMAP data and our approach to detecting
melt events using SMAP microwave observations. In Sect. 3, we introduce our geophysical forward modeling, and Sect. 4
explains our empirical and model-based melt detection and snow wetness retrieval algorithms. Sect. 5 demonstrates the melt
detection and snow wetness retrieval results from both the empirical and the model-based algorithms. Sect. 6 presents our
conclusions.


## 2    Data and Methods

### 2.1    SMAP Data

NASA launched the SMAP mission on January 31, 2015; the science data production began on March 31, 2015. The L-band
radiometer on-board the satellite includes vertically (V) and horizontally (H) polarized brightness temperature (TB) channels.
The SMAP TB measurements have a 38 km spatial footprint (defined by the half-power footprint on the Earth's surface of the
radiometer antenna pattern), and the data are gridded on a 9-km polar equal-area projection grid (Chaubell, et al., 2016). The





SMAP satellite has a sun-synchronous 6AM/6PM equator-crossing orbit, a constant 40° sensor incidence angle, and an approximate 1000-km swath width (Piepmeier et al., 2017). This enables daily coverage of the Antarctic Ice Sheet with both AM and PM overpasses. The radiometric resolution of the gridded SMAP TB product is less than 0.5 K (Piepmeier et al., 2017).

## 2.2  Method

The presence of even a small amount of liquid water in the surface snowpack significantly impacts the electrical properties of snow at microwave frequencies (Ulaby & Long, 2014). This results in large changes in the microwave TB measurements, which permits melt detection and derivation of melt related characteristics (Ulaby & Long, 2014). Our method is composed of detecting melt events from the changes of normalized polarization ratio (NPR) and V-polarized TB with respect to reference values computed during winter conditions, and subsequently, estimating snow wetness for melt events using a snow model. We use snow wetness as an indicator of melt intensity. The NPR is computed as follows:

$$\mathrm{NPR} = \frac{T_{B_v} - T_{B_h}}{T_{B_v} + T_{B_h}} \tag{1}$$

where $T_{B_v}$ and $T_{B_h}$ are V- and H-polarized TB, respectively. The advantage of using NPR in addition to $T_{B_v}$ is that it does not depend on the physical temperature of the snow and ice, but is a function of dielectric changes, which vary between different seasons.

Fig. 1 (a) and (b) show the regional pattern of the NPR reference value (NPR$_{\mathrm{ref}}$) and the maximum NPR seasonal difference from reference conditions for the 2015-2016 austral melt season (Oct 31, 2015 – May 31, 2016), respectively. The NPR$_{\mathrm{ref}}$ is the temporal mean of the *NPR* from Oct 17 to Oct 31 in each calendar year. The analysis showed that the NPR value both increased and decreased from the winter season average at different locations. The positive NPR change is expected for dry snow conditions or areas that typically experience limited seasonal melting. These areas consist of layered snow and firn, and penetration depths can be up to hundreds of meters (Jezek et al., 2015). The NPR is relatively low in these areas, and Melt events cause the NPR to increase because the presence of meltwater causes a greater V and H polarization difference than under dry snow/ice conditions. The case for the negative NPR change is somewhat more complex. In areas that experience seasonal melting with complex subsurface structures like ice pipes and lenses, the penetration depth is reduced to tens of meters (Miller et al., 2020). The NPR is relatively high for these areas, and the presence of meltwater during the melt events causes the NPR to decrease because it extinguishes signals with a large polarization difference emanating from structures inside the ice sheet.

After detecting the melt areas, we retrieve snow wetness based on a geophysical multi-layer snow emission model. The model relates observed TB changes to the amount of liquid water in the surface layers of the ice sheet. The model is formulated somewhat differently for the increasing and decreasing NPR cases as is detailed in Sect. 3.





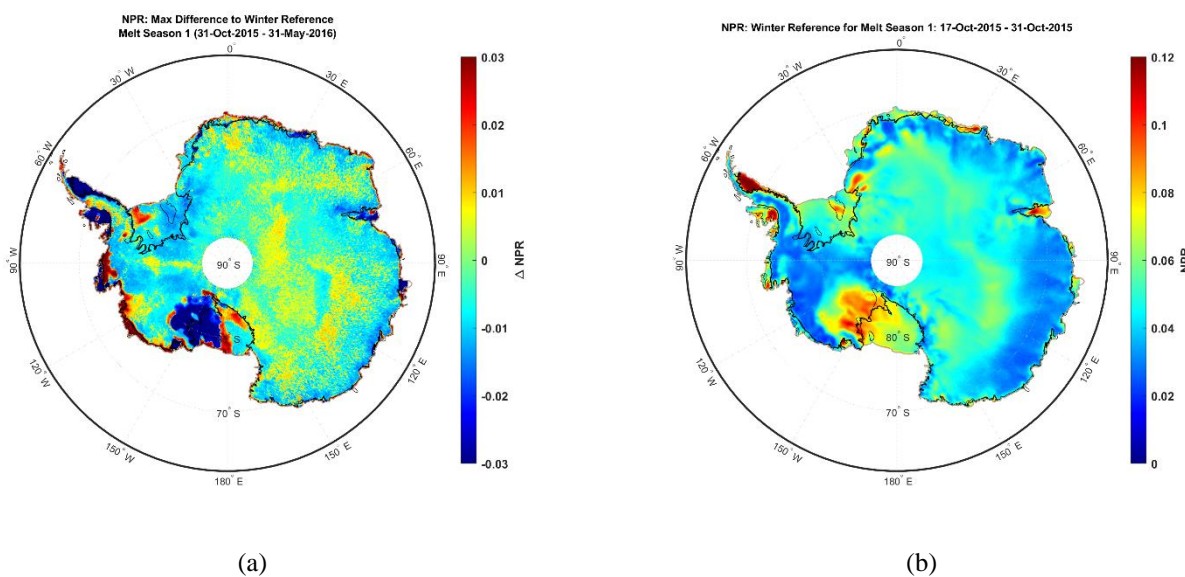

|(a)|(b)|
|---|---|

**Fig. 1. (a) The maximum of the SMAP $\Delta NPR$ and (b) $NPR_{ref}$ for an austral melt season (Oct 31, 2015 – May 31, 2016) over Antarctica.**

## 3  Forward Geophysical Modelling

In this section, we calculate the H- and V-polarization TB ($\mathbb{T}_{Bh}$, $\mathbb{T}_{Bv}$) for a multi-layer medium using the incoherent approach of radiative transfer (RT) theory (Ulaby & Long, 2014; Tsang, et al., 2000). As the $NPR$ increases and decreases during the melt season depending on the location, we developed two separate models for each of these scenarios. One scenario has a three-layer medium (air, wet snow, and dry snow layers) for the case of increasing NPR, and one with a four-layer medium by adding a middle layer between the wet and dry snow layers of the three-layer model for the case of a decreasing NPR.

### 3.1    Three-Layer Model (Increasing NPR)

Fig. 2(a) shows the configuration of the three-layer model, which consists of air, wet snow, and semi-infinite dry snow layers with two boundaries at $z = -d_1 = 0$ and $z = -d_2 = -d_{wet2}$, where $d_{wet2}$ is the thickness of the wet snow layer. In this schematic diagram, $\theta_2$ and $\theta_3$ are the angles of the wave propagation inside layers 2 and 3, respectively, which can be found from the known observation angle, $\theta_1$, using Snell's law. The dielectric constant of the wet snow and dry snow layers are

estimated as explained in (Ulaby & Long, 2014).

Because of the long wavelength of L-band, the model does not include volume and surface scattering, which are critically important at higher frequencies. For simplicity, the model assumes ideal conditions for the environment, and therefore ignores other complicating factors such as radio frequency interference and atmospheric attenuation. The upward and downward traveling components of the TB in the $n$-th layer are given by


$$\mathbb{T}_{B_{2p}}^u(\theta_2, z) = \mathbb{T}_{B_{2p}}^u(\theta_2, -d_2)e^{-k_{a_2}(z+d_2)\sec\theta_2} + \left(1 - e^{-k_{a_2}(z+d_2)\sec\theta_2}\right)T_{0_2} \qquad (2)$$





$$\mathbb{T}^d_{B_{2p}}(\theta_2, z) = \mathbb{T}^d_{B_{2p}}(\theta_2, -d_1)e^{k_{a_2}(z+d_1)\sec\theta_2} + \left(1 - e^{k_{a_2}(z+d_1)\sec\theta_2}\right)T_{0_2} \tag{3}$$

where the subscript $p = (h, v)$ stands for the polarization, $T_{0_2}$ is the physical temperature of the second layer, and $k_{a_2} = -2\Im\{\omega\sqrt{\mu_2\epsilon_2}\}$ is the power absorption coefficient of the layer 2, where $\mu_2 = \mu_1$ (nonmagnetic material), and $\epsilon_2 = \epsilon'_2 - j\epsilon''_2$.

The superscripts $u$ and $d$ show upward and downward dwelling components of the TB. The boundary conditions at the top and lower boundary are given by

$$\mathbb{T}^d_{B_{2p}}(\theta_2, z = -d_1) = \Gamma_{p_{12}}\mathbb{T}^u_{B_{2p}}(\theta_2, z = -d_1) \tag{4}$$

$$\mathbb{T}^u_{B_{2p}}(\theta_2, -d_2) = \Gamma_{p_{23}}\mathbb{T}^d_{B_{2p}}(\theta_2, -d_2) + \left(1 - \Gamma_{p_{23}}\right)T_{0_3} \tag{5}$$

where $T_{0_3}$ is the physical temperature of the third layer, and $\Gamma_{p_{12}}$ and $\Gamma_{23}$ are the Fresnel reflectivity at the top and lower boundaries, respectively. It is assumed that there is no horizontal variation, and the boundaries are locally flat within the

antenna footprint. Using equations (2)-(5), the upward and downward emission are given by

$$\begin{aligned}
\mathbb{T}^d_{B_{2p}}(\theta_2, z) = \frac{\Gamma_{p_{12}}\Gamma_{p_{23}}e^{k_{a_2}(z-d_2)\sec\theta_2}}{1 - \Gamma_{p_{12}}\Gamma_{p_{23}}e^{-2k_{a_2}d_2\sec\theta_2}}\Big[&\Gamma_{p_{12}}\left(1 - \Gamma_{p_{23}}\right)T_{0_3}e^{-2k_{a_2}d_2\sec\theta_2}\\
&+ \Gamma_{p_{12}}\left(e^{-k_{a_2}d_2\sec\theta_2} - e^{-2k_{a_2}d_2\sec\theta_2}\right)T_{0_2} + \left(1 - e^{-k_{a_2}d_2\sec\theta_2}\right)T_{0_2}\Big]\\
&+ \Gamma_{p_{12}}\left(1 - \Gamma_{p_{23}}\right)T_{0_3}e^{k_{a_2}(z-d_2)\sec\theta_2} + \Gamma_{p_{12}}\left(e^{k_{a_2}z\sec\theta_2} - e^{k_{a_2}(z-d_2)\sec\theta_2}\right)T_{0_2}\\
&+ \left(1 - e^{-k_{a_2}d_2\sec\theta_2}\right)T_{0_2}
\end{aligned} \tag{6}$$

$$\mathbb{T}^u_{B_{2p}}(\theta_2, z) = \Gamma_{p_{23}}\mathbb{T}^d_{B_{2p}}(\theta_2, -d_2)e^{-k_{a_1}(z+d_2)\sec\theta_2} + \left(1 - \Gamma_{p_{23}}\right)T_{0_3}e^{-k_{a_2}(z+d_2)\sec\theta_2} + \left(1 - e^{-k_{a_2}(z+d_2)\sec\theta_2}\right)T_{0_3} \tag{7}$$

Finally, the estimated TB at the radiometer antenna ($\mathbb{T}_{Bp}$) is given by (8), which is approximately equal to the TB just above the top boundary (air-snow) as the atmospheric attenuation is assumed to be negligible in the SMAP L-band frequency (1.41

GHz).

$$\mathbb{T}_{Bp}(\theta_1) \approx \mathbb{T}^u_{B_{1p}}(\theta_1, 0) = \left[1 - \Gamma^p_{12}(\theta_1)\right]\mathbb{T}^u_{B_{2p}}(\theta_2, 0) \tag{8}$$



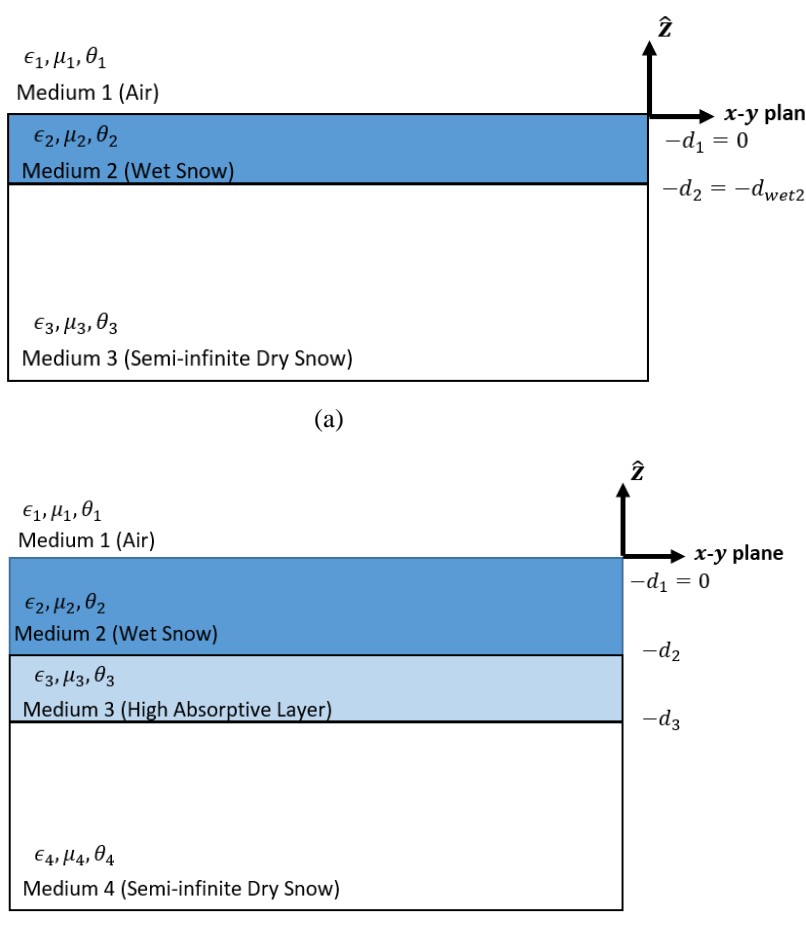

(a)

(b)

**Fig. 2. Configuration of the (a) three-layer medium and (b) four-layer medium model design.**

### 3.2  Four-Layer Model (Decreasing NPR)

Fig. 2(b) shows the configuration of the four-layer model, which consists of air, wet snow, a high absorptive layer, and a semi-
infinite dry snow layers with boundaries at $z = -d_1 = 0$ and $z = -d_2 = -d_{wet2}$, and $z = -d_3 = -(d_{wet2} + d'_3)$ where
$d_{wet2}$ and $d'_3$ are the thicknesses of the layers 2 and 3. As the medium 3 does not fall into the category of the wet snow or the
dry snow models, its real and imaginary parts are considered as two independent parameters in our model.

To streamline the TB estimation, an effective reflectivity $\Gamma_{p_{eff}}(\theta_2)$ and effective physical temperature $T_{0_{eff}}$ are used for a
composite medium combining layer 3 and 4 similar to (Tan, et al., 2019). Then, after introducing the top layer 2, the boundary
conditions are given by





$$\mathbb{T}^d_{B_{2p}}(\theta_n, z = -d_1) = \Gamma_{p_{12}} \mathbb{T}^u_{B_{2p}}(\theta_2, z = -d_1) \tag{9}$$

$$\mathbb{T}^u_{B_{2p}}(\theta_2, -d_2) = \Gamma_{p_{eff}}(\theta_2) \mathbb{T}^d_{B_{2p}}(\theta_2, -d_2) + \left(1 - \Gamma_{p_{eff}}(\theta_2)\right) T_{0_{eff}} \tag{10}$$

The $\Gamma_{p_{eff}}(\theta_2)$ is equal to the Fresnel reflectivity of layered media with electrically smooth boundaries, which also consists of the coherent interactions, and $T_{0_{eff}}$ is related to the TB of the composite medium of layers 3 and 4, $\mathbb{T}_{B_{comp}}(\theta_2)$, as given by

(11). The $\mathbb{T}_{B_{comp}}(\theta_2)$ for this composite medium of layers 3 and 4 can be found using equations (6)-(8), as given in the Sect. 3.1.

$$T_{0_{eff}} = \frac{\mathbb{T}_{B_{comp}}(\theta_2)}{1 - \Gamma_{p_{eff}}(\theta_2)} \tag{11}$$

Finally, the observed TB can be calculated using equations (6)-(8) similar to Sect. 3.1, as it is now a three-layer medium with the above effective reflectivity and temperature.

### 3.3 Simulations

Figs. 3(a) and (b) illustrate an example where the simulated TB decreases and the $NPR$ increases with increasing snow wetness (three-layer model). Table 1 shows the layer properties for this three-layer model. In Figs. 4(a) and (b) the simulated TB increases and the $NPR$ decreases with increasing snow wetness (four-layer model). Table 2 shows the layer properties for this four-layer model.

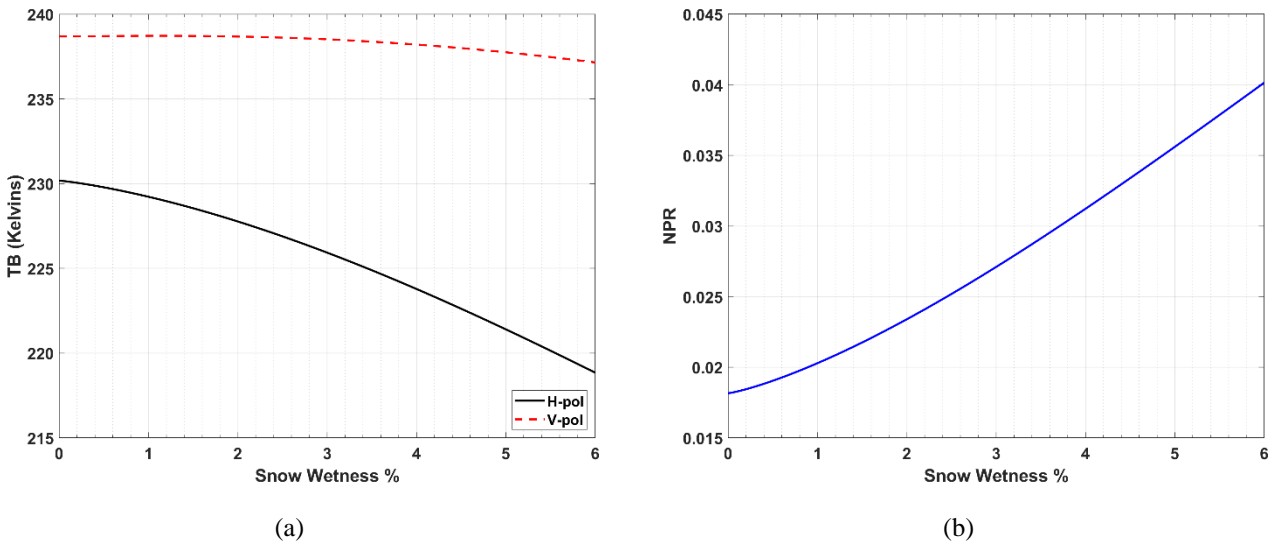

(a)                                                                 (b)

**Fig. 3. (a) Estimated brightness temperature (TB) and (b) normalized polarization ratio (NPR) changes with snow wetness as derived**
**from the three-layer model.**



**Table 1. Layer Properties for the three-layer model.**

| Layer | Density | Thickness | Physical Temperature | Dielectric Constant |
|---|---|---|---|---|
| Medium 2 (Wet Snow) | 450 kg/m³ | $d_2 - d_1 = 3$ cm | 273 K | From (Ulaby & Long, 2014) |
| Medium 3 (Semi-infinite Dry Snow) | 450 kg/m³ | *Not Required* | 240 K | From (Ulaby & Long, 2014) |

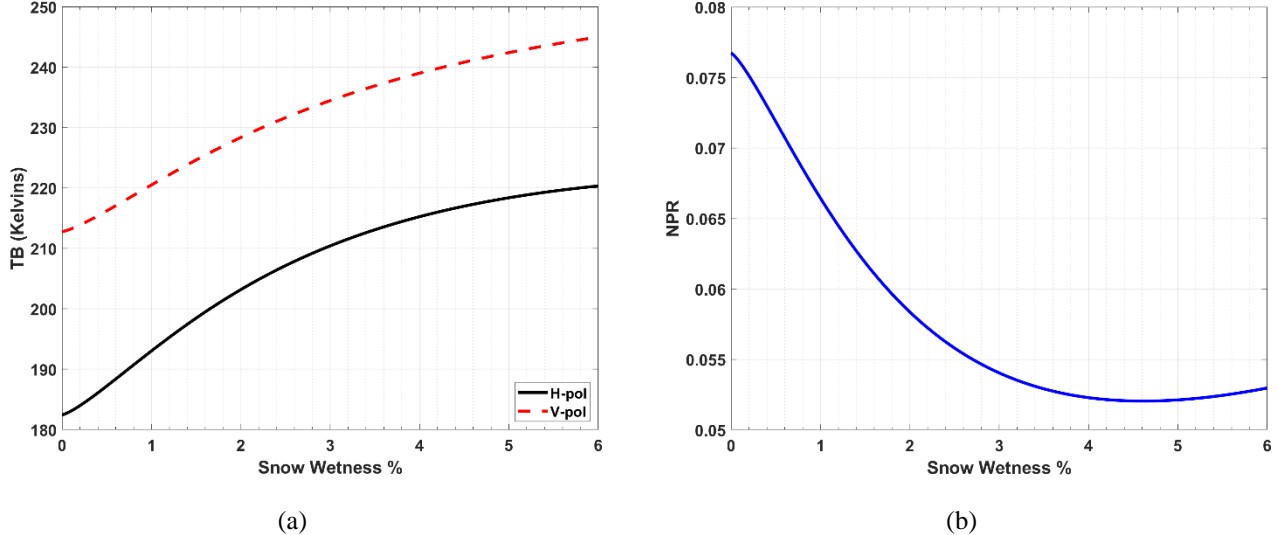

(a)                                   (b)

**Fig. 4. (a) Estimated brightness temperature (TB) and (b) normalized polarization ratio (NPR) changes with snow wetness as derived from the four-layer model.**

**Table 2. Layer properties for the four-layer model.**

| Layer | Density | Thickness | Physical Temperature | Dielectric Constant |
|---|---|---|---|---|
| Medium 2 (Wet Snow) | 450 kg/m³ | $d_2 - d_1 = 25$ cm | 273 K | From (Ulaby & Long, 2014) |
| Medium 3 (High Absorptive Layer) | *Not Required* | $d_3 - d_2 = 50$ cm | 270 K | $3.5 - 9j$ |
| Medium 4 (Semi-infinite Dry Snow) | 450 kg/m³ | *Not Required* | 240 K | From (Ulaby & Long, 2014) |



## 4 Melt Detection and Snow Wetness Retrieval Algorithm

### 4.1 Empirical Threshold Algorithm

An empirical threshold algorithm is used to detect melt events. The algorithm determines that a melt event has occurred if both $\Delta NPR$ ($= NPR_{daily} - NPR_{ref}$) and $\Delta T_{BV}$ ($= T_{BV\,daily} - T_{BV\,ref}$) are greater than an empirically found threshold value, as given by (12) and (13), respectively.

$$m_1(t) = \begin{cases} 1 \ (True) & |\Delta NPR| \geq Z_{npr} \times E[SD[NPR]|_{WREF}]_{All\ Pixels} \\ 0 \ (False) & otherwise \end{cases} \tag{12}$$

$$m_2(t) = \begin{cases} 1 \ (True) & |\Delta T_{BV}| \geq Z_{tbv} \times E[SD[T_{BV}]|_{WREF}]_{All\ Pixels} \\ 0 \ (False) & otherwise \end{cases} \tag{13}$$

where E[] stands for the mean estimator; SD[] stands for the temporal standard deviation estimator; WREF refers to the time period from Oct 17 to Oct 31; All Pixels refers to taking the spatial average over all Antarctic pixels; and $Z_{npr}$ and $Z_{tbv}$ are constant real numbers. This formulation relates the threshold to the variance of the NPR and $T_{BV}$, while $Z_{npr}$ and $Z_{tbv}$ are used as tuning parameters to determine appropriate threshold levels for each grid cell. A melt event will be detected at time $t$ if both $m_1(t)$ and $m_2(t)$ indicate a melt event, which corresponds to a bitwise AND operation ($\wedge$) on $m_1(t)$ and $m_2(t)$ binary states.

A logic truth table is shown in Table 3, where $m(t)$ is a dimensionless binary state variable designating melt (1) and frozen (0) conditions.

The Z parameters dictate how much the $\Delta NPR$ and $\Delta T_{BV}$ need to deviate from the reference level in order to result in a positive indication for melt. Conversely, the Z parameter determines the false alarm rate (FAR) which can be defined as (De Roo, et al., 2007; Mousavi, et al., 2018):

$$FAR = \frac{1}{2}\left(1 - \text{erf}\left(Z/\sqrt{2}\right)\right) \tag{14}$$


where $\text{erf}(Z)$ is the error function (Mousavi, et al., 2018). For example, the FAR will be about 2.2% and 15.8% for $Z = 2$ and $Z = 1$, respectively. Since the FAR for each day in a melt season is independent of other days, the FAR for all days in a melt season is related to the FAR for a single day, as given by

$$1 - FAR(\text{for all days in a melt season}) = \left(1 - FAR(\text{for one day})\right)^{n_{ms}} \tag{15}$$

where $n_{ms}$ is the total number of days in a melt season, which is from Oct 31 of each calendar year to the May 31 of the following year, and $n_{ms} = 212$ (assuming February is 28 days). Because we want to keep the FAR small, it can be assumed that Z is selected such that FAR $\ll 1$. Therefore, (15) can be simplified to



$$\text{FAR(for all days in a melt season)} = n_{ms}\ \text{FAR(for one day)} = \frac{n_{ms}}{2}\left(1 - \text{erf}\left(Z/\sqrt{2}\right)\right) \tag{16}$$

Since the melt detection is based on a bitwise AND operation, the lower value of the $Z$ parameter will dictate the final FAR.
Using these conditions for the $NPR$ and $T_{BV}$, the right value for the minimum number of days for the winter reference, $N_{ref}$, and the $Z$ parameters can be found by simultaneously satisfying (17a) and (17b).

$$N_{ref} > \left(\frac{Z_{npr}}{\max\{\Delta NPR\}}\right)^2 \times \sum_{i=1}^{N_{ref}} \left(NPR_i - NPR_{ref}\right)^2 \tag{17a}$$

$$N_{ref} > \left(\frac{Z_{tbv}}{\max\{\Delta T_{BV}\}}\right)^2 \times \sum_{i=1}^{N_{ref}} \left(T_{BV\,i} - T_{BV\,ref}\right)^2 \tag{17b}$$

A spatial average of the $N_{ref}$ over all pixels can be performed to find a unique and fixed number for the required winter reference days for a given $Z$ parameter. As an example, for the 2015-2016 austral melt season, Figs. Fig. 5(a) and (b) show the region of values that make the inequalities in (17a) and (17b) true (shaded green), respectively. The false region is shown in red. It can be observed that a higher $Z$ value would require a higher $N_{ref}$ value. Even though choosing a higher $Z$ value will decrease the FAR, it will decrease the number of melt days, which results in missing days. Our proposed two week interval from Oct 17-31 for the winter reference period has enough samples to satisfy these conditions with $Z_{NPR} = 5$ and $Z_{T_{BV}} = 10$. The threshold values for $|\Delta NPR|$ and $|\Delta T_{BV}|$ using these proposed $Z$ parameters are 0.010-0.011 and 6.97-7.31 K, respectively, for all the austral melt seasons from 2015 to 2020.

**Table 3. Logic truth table for the $m(t)$ dimensionless binary state variable. Green is for logic 1 (True), and red is for logic 0 (False).**

| $m_1(t)$ | $m_2(t)$ | $m(t) = m_1(t) \wedge m_2(t)$ |
|---|---|---|
| 1 (True) | 1 (True) | 1 (Melt) |
| 1 (True) | 0 (False) | 0 (Frozen) |
| 0 (False) | 1 (True) | 0 (Frozen) |
| 0 (False) | 0 (False) | 0 (Frozen) |





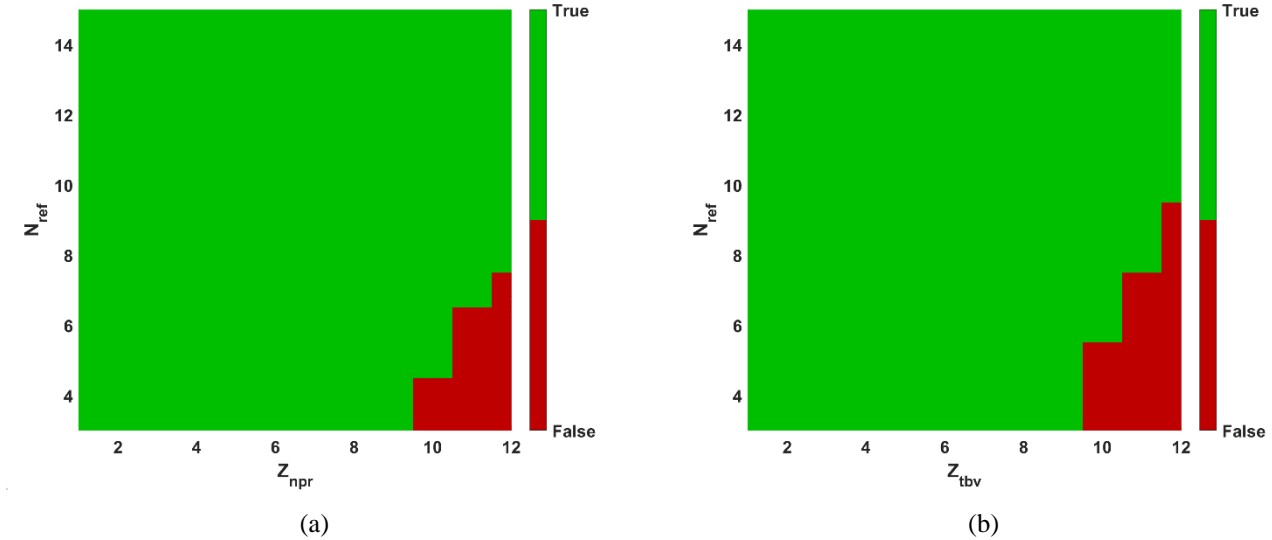

(a)                          (b)

**Fig. 5. The region of values that make the inequalities in (a) (17a) and (b) (17b) true (green) and false (red) during the 2015-2016 austral melt season.**

The main purpose of monitoring $T_{BV}$ is to avoid false melt detection, because the $NPR$ may change in some cases due to the changes in the snow/ice vertical structure rather than due to a melt event (because $T_{BH}$ is more sensitive to the vertical structure changes than $T_{BV}$), while $T_{BV}$ mainly changes from snow wetness and temperature variations. As an example, Figs. 6 and 7 show $T_{BV}$ and $NPR$, respectively, measured by SMAP over the Southern George VI ice shelf ($71.31^o$ S, $68.32^o$ W) during the first austral melt season (Oct 31, 2015 - May 31, 2016). The figures also show in-situ air temperature measurements from

the Fossil Bluff site located at the George VI sound. Around January 2016, there is a significant change in the measured $NPR$ (decrease) and $T_{BV}$ (increase), and they both fall outside their corresponding thresholds (cyan shaded region), indicating melt detection. The SMAP derived melt detection also coincides with a substantial increase in surface air temperatures to within ±5ºC, indicating conditions conducive to melt.

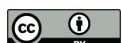



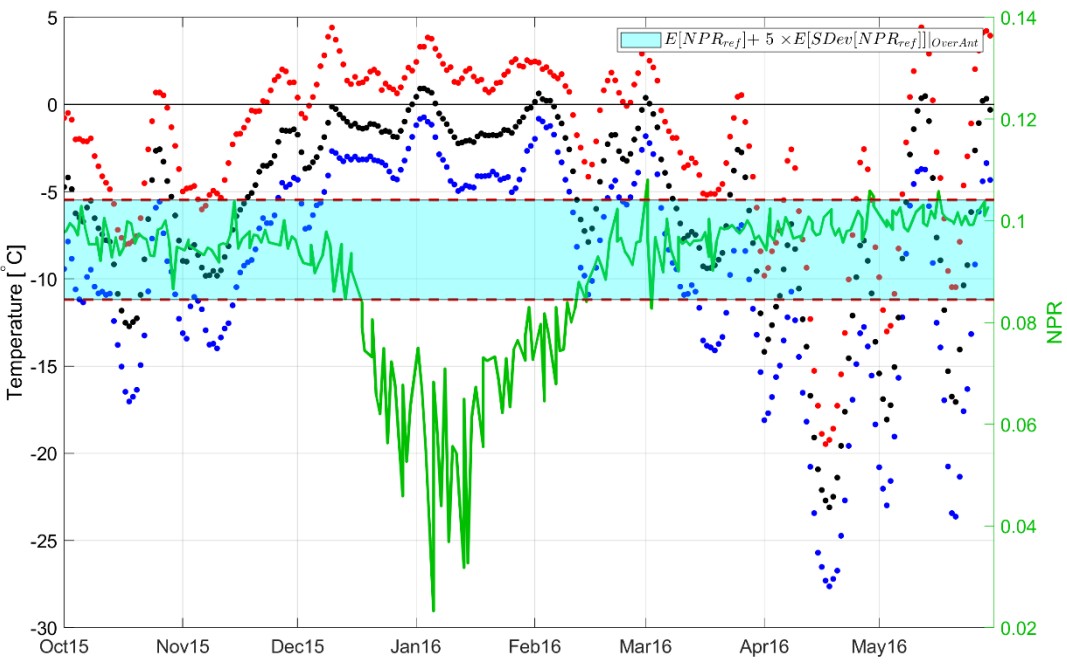

**Fig. 6. Measured *NPR* by SMAP for the Southern George VI ice shelf (71.31° S, 68.32° W) during the first austral melt season (Oct 31, 2015 - May 31, 2016) of the study period. The temperature is measured at the Fossil Bluff station. Daily maximum, minimum and average air temperatures are denoted by respective red, blue and black dots.**



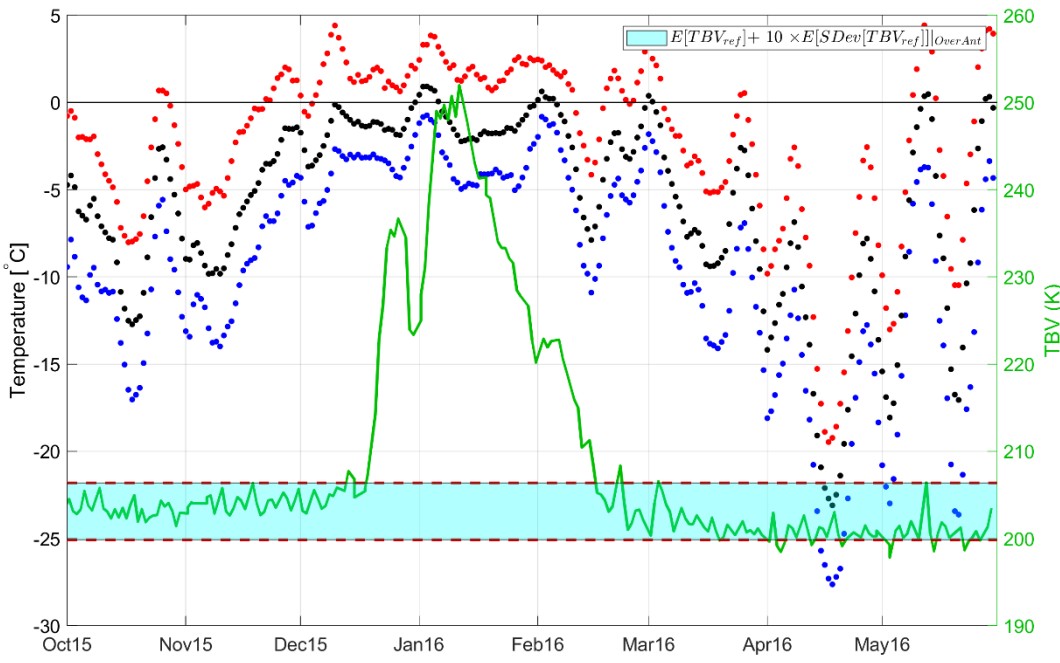
**220** **Fig. 7. Measured V-pol brightness temperature by SMAP for the Southern George VI ice shelf ($71.31^o$ S, $68.32^o$ W) during the first austral melt season (Oct 31, 2015 - May 31, 2016). The temperature is measured at the Fossil Bluff station. Daily maximum, minimum and average air temperatures are denoted by respective red, blue and black dots.**

### 4.2 Model Based Snow Wetness Retrieval (SnoWR) Algorithm

The snow wetness retrieval (SnoWR) algorithm uses the microwave emission models explained in Sect. 3 for either decrease

**225** (NPR-DECR) or increase (NPR-INCR) scenarios of the $NPR$ during the melt season (MS). Four different look-up-tables (LUTs) were made by sweeping the model parameters over specified realistic ranges. Separate LUTs are computed for the frozen (LUT-FS-INCR) and melt (LUT-MS-INCR) seasons for both NPR-INCR and NPR-DECR cases. The LUT-MS-INCR uses the three-layer model, while the LUT-FS-INCR uses a two-layer model where the wet snow layer (medium 2) in Fig. 2(a) is entirely absent during the frozen season. The LUT-MS-DECR uses the four-layer model, while the LUT-FS-DECR uses a

**230** customized three-layer model where the wet snow layer (medium 2) in Fig. 2(b) is absent. The high-absorptive layer is assumed to remain unchanged between the frozen and melt seasons.

The algorithm first separates the pixels for NPR-INCR and NPR-DECR scenarios using the statistics of $NPR$ and $T_{BV}$ and equation (12) and (13) in Sect. 4.1. These conditions should be evaluated without their absolute values so that the positive and negative changes can separate NPR-INCR and NPR-DECR pixels. In NPR-INCR pixels, $NPR$ and $T_{BV}$ change in positive and

**235** negative directions, respectively, while in NPR-DECR pixels, $NPR$ and $T_{BV}$ change in negative and positive directions. After selecting and separating the melt pixels for the NPR-INCR and NPR-DECR cases, the algorithm starts to estimate the layer




properties, such as thickness and dielectric constant, in each case separately, by comparing the LUT and measured $T_{BH}$ and $T_{BV}$ values.

In the case of NPR-INCR during the FS, by comparing the LUT-FS-INCR with the temporal maximum of the measured $T_{BV}$,

the density and temperature of the medium 4 is retrieved. Only $T_{BV}$ is used here, as there is no vertical layer structure during the FS in the dry snow layers. During the MS, LUT-MS-INCR is compared with the maximum of $T_{BH}$ and $T_{BV}$, as the snowpack starts to have vertical layer structures. Hence, the properties of the wet snow layer are estimated by averaging their retrieved values from tuning for only $T_{BH}$ and $T_{BV}$ cases; the estimated properties include snow/ice density, thickness, and maximum wetness percentage ($m_{v2_{max}}$).

Next, another LUT is made by sweeping the wetness range of the wet layer ($m_{v2}$) over $[0 \ m_{v2_{max}}]$. The resulting LUT is used to derive two different daily snow wetness values ($m_{v2_{T_{BH}}}$, $m_{v2_{T_{BV}}}$) by minimizing the difference between the LUT and SMAP observed $T_{BH}$ and $T_{BV}$ values. Using $m_{v2_{T_{BH}}}$ and $m_{v2_{T_{BV}}}$, two different sets of daily brightness temperatures can be simulated. By comparing the temporal mean of the error between the LUT and SMAP values, the best snow wetness value can be retrieved as follows:

$$m_{v2_{daily}} = \begin{cases} m_{v2TBV} & if \ ERR_{TBV}(m_{v2TBV}) < \ ERR_{TBH}(m_{v2TBV}) \\ m_{v2TBH} & o.w \end{cases} \quad (18)$$


where $ERR_{TBV}(m_{v2TBV})$ is the temporal mean of the estimated $T_{BV}$ using $m_{v2TBV}$, and $ERR_{TBH}(m_{v2TBV})$ is the temporal mean of the estimated $T_{BH}$ using $m_{v2TBV}$. This step will correct for any possible bias in the modeled TB, and may be excluded if there is minimal bias in the measured values.

After retrieving the daily snow wetness values ($m_{v2_{daily}}$), the reliability of the retrieval can be assessed by simulating TB

using the retrieved $m_{v2_{daily}}$.

Using a threshold value $m_{v2TH} = m_{v2_{ref}} + Z_{m_v} \times \sqrt{Var[m_{v2_{ref}}]}$, the corrected retrieved snow wetness, $m_{v2_{daily}}^{COR}$, is given by equation (19). The parameter $Z_{m_v}$ determines an independent FAR using equation (14).

$$m_{v2_{daily}}^{COR} = \begin{cases} 0 & m_{v2_{daily}} \leq m_{v2TH} \\ m_{v2_{daily}} - m_{v2TH} & m_{v2_{daily}} > m_{v2TH} \end{cases} \quad (19)$$

In the case of NPR-DECR, daily snow wetness ($m_{v2_{daily}}$) is retrieved, and $T_{BH}$ and $T_{BV}$ simulated similar to the NPR-INCR

case with two differences. First, the temporal minimum and maximum of the measured TB are used during the FS and MS, respectively, as the TB increases during the MS in this scenario. Second, both $T_{BV}$ and $T_{BH}$ are used in the tuning as the NPR-DECR pixels have vertical layer structure in both FS and MS. The remaining steps are similar to the NPR-INCR scenario. Fig. 8 shows the SnoWR algorithm flowchart.



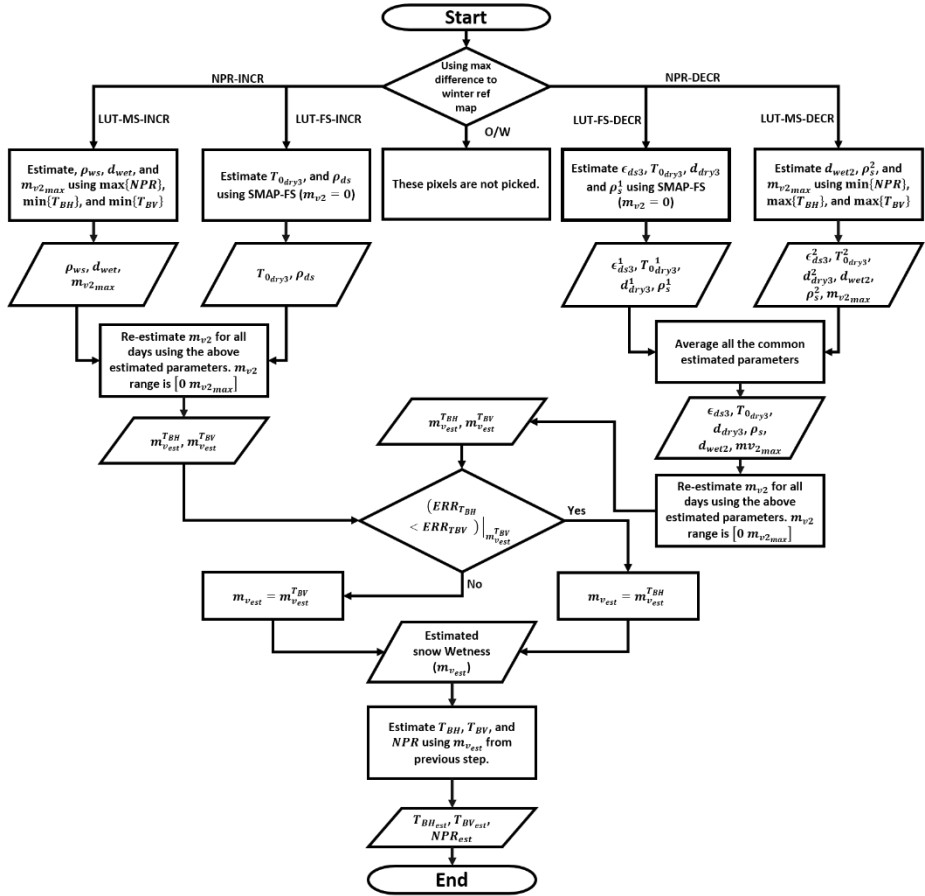


**Fig. 8. Processing flow of the SMAP TB snow wetness retrieval (SnoWR) algorithm. The main output is $m_{v_{est}}$.**

As an example, Figs. 9 and 10 show the measured and simulated $T_{BH}$ and $T_{BV}$, respectively, for the Wilkins Ice Shelf (70.25$^o$

S, 73.00$^o$ W) during 2015-2020. The simulated $T_{BV}$ is better matched to the measured values (Fig. 10) compared to $T_{BH}$ (Fig.

9) because $m_{v2_{daily}} = m_{v2_{TBV}}$, see (18). The error values are $ERR_{T_{BV}}(m_{v2_{TBV}}) = 0.58$ K and $ERR_{T_{BH}}(m_{v2_{TBV}}) = 12.62$ K.

The snow wetness variation is well retrieved during different seasons, as shown in Fig. 11. The estimated snow wetness values

are based on the model physical parameters used to match with the SMAP measurements. Because the tuning process includes

an empirical adjustment of the layer parameters and in-situ snow wetness values are unavailable for an assessment and

comparison, a quantitative accuracy measure of the snow wetness retrieval is not reported in this paper. However, the estimated

snow wetness range is similar to measured values reported from previous studies (Willatt, et al., 2010). The retrieved snow

wetness in Fig. 11 is the corrected snow wetness as given by (19) for $Z_{m_v} = 2$.



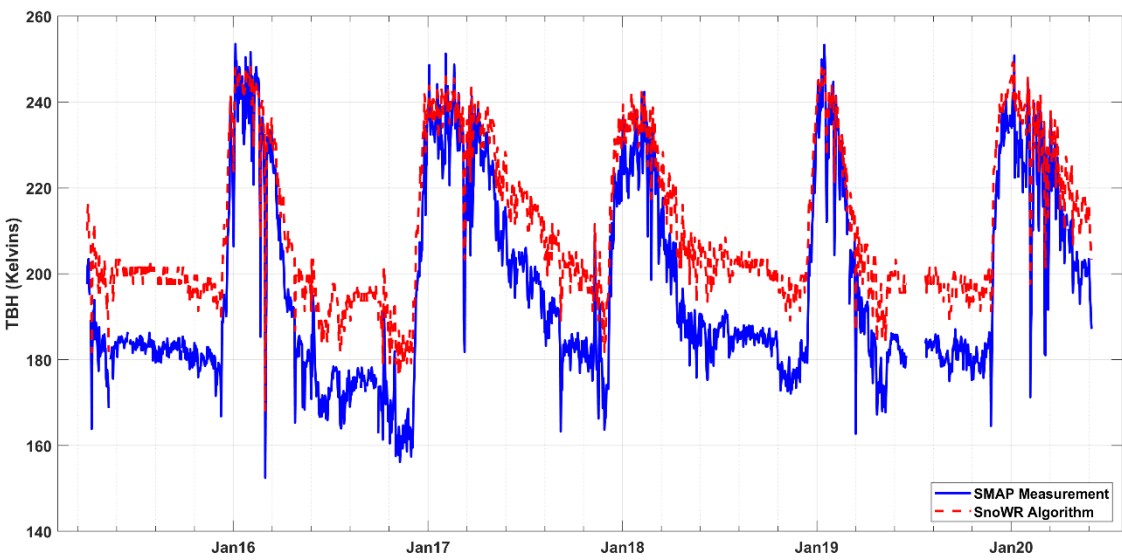

**Fig. 9. The estimated TBH from the SnoWR algorithm (red dashed line) and measured by SMAP (blue solid line) for the Wilkins ice shelf ($70.25^o$ S, $73.00^o$ W) as a function of time (2015-2020).**


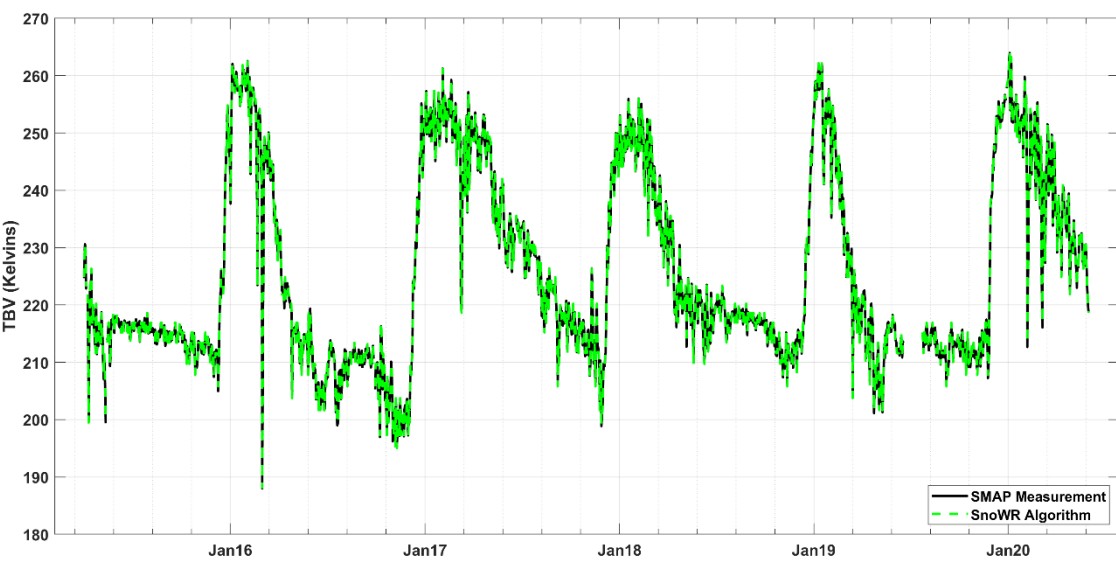

**Fig. 10. The estimated TBV from the SnoWR algorithm (green dashed line) and measured by SMAP (black solid line) for the Wilkins ice shelf ($70.25^o$ S, $73.00^o$ W) as a function of time (2015-2020).**






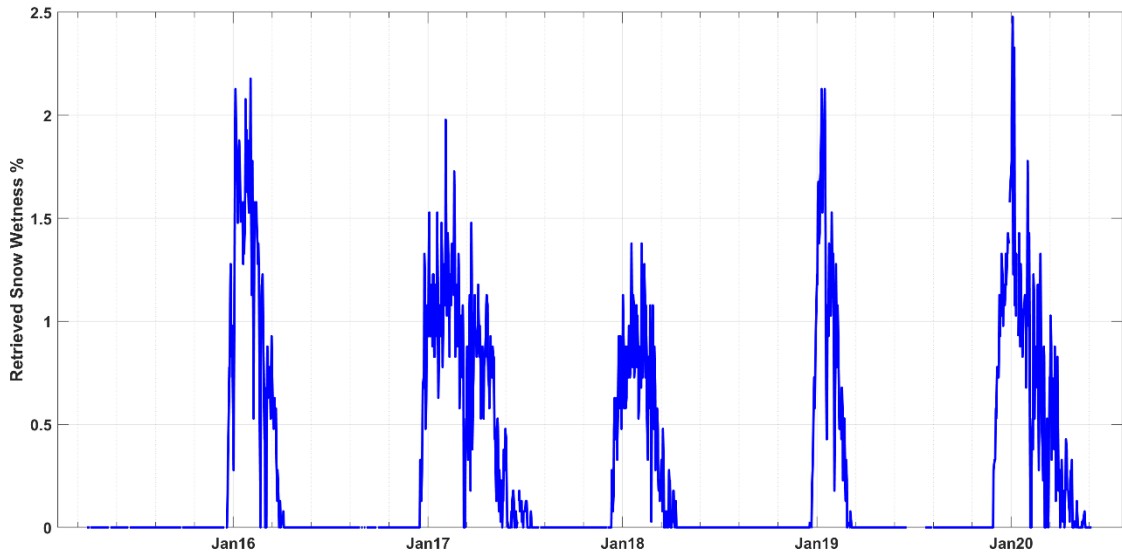

**Fig. 11. The estimated snow wetness percentage from the SnoWR algorithm for the Wilkins ice shelf ($70.25^o$ S, $73.00^o$ W) as a function of time (2015-2020).**

## 5    Results

### 5.1    Empirical Threshold Algorithm Results

Fig. 12 shows the most recent Antarctica digital elevation model (DEM) with various ice shelves denoted (Howat, et al., 2019). The Antarctica ice mask used in this study is obtained from the Moderate Resolution Imaging Spectroradiometer (MODIS) for the 2013-2104 (Scambos, et al., 2007; Haran, et al., 2018, updated 2019). This ice mask matches the outline of the Antarctica DEM in Fig. 12. Table 4 shows the melt seasons and winter reference periods used in this study. The potential melt seasons start on the last day of their winter reference period.


**Table 4. The austral melt seasons and their corresponding proposed winter references periods.**

| Melt Season No. | Melt Season Period | Winter Reference Period |
|---|---|---|
| 1 | Oct 31, 2015 - May 31, 2016 | Oct 17, 2015 - Oct 31, 2015 |
| 2 | Oct 31, 2016 - May 31, 2017 | Oct 17, 2016 - Oct 31, 2016 |
| 3 | Oct 31, 2017 - May 31, 2018 | Oct 17, 2017 - Oct 31, 2017 |
| 4 | Oct 31, 2018 - May 31, 2019 | Oct 17, 2018 - Oct 31, 2018 |
| 5 | Oct 31, 2019 - May 31, 2020 | Oct 17, 2019 - Oct 31, 2019 |



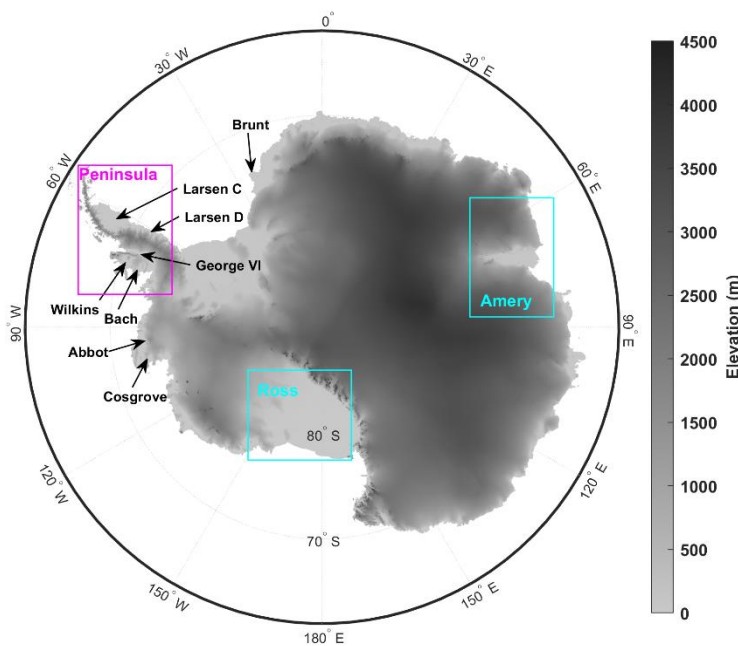

**Fig. 12. Antarctica digital elevation model.**

300 Fig. A1 in Appendix A shows the maximum $\Delta NPR$ and $NPR_{ref}$ measured by the SMAP L-band microwave radiometer for each austral melt season. Similarly, Fig. A2 shows the $\Delta T_{BV}$ and $T_{BV\,ref}$. Melt events are then detected by using the empirical melt detection threshold given in equations (12) and (13) and Table 3. Fig. A1 and Fig. A2 show that there is very little inter-annual variability in the $NPR_{ref}$ and $T_{BV_{ref}}$ values.

Fig. 13 shows the melt detection results overlaid on the Antarctica DEM. The maps show that MS 1 overall experienced the

305 largest melt extent as a result of the exceptional Ross Ice Shelf (81.50° S, 175.00° W) melt event (Nicolas, et al., 2017). A narrow strip along the Transantarctic mountains to the east of the Ross Ice Shelf also experienced melt events in MS 2. The Brunt ice shelf (75.55° S, 25.00° W) experienced melt events during all melt seasons except MS 1. Ice shelves along periphery of the Antarctic Peninsula (Larsen C, Wilkins, George VI, Bach) experienced consistent melt events during each MS. In addition, ice shelves along the Amundsen-Bellingshausen Sea coast (~90.00° W), such as Abbot and Cosgrove, experienced

310 melt events in MS 1 and 5.

The Ross Ice Shelf experienced a short period of melt (~14 days) during MS 1, while the ice shelves along the periphery of the Antarctic Peninsula experienced about 40 melt days on average between MS 1 and 5 and a long-duration of melt in MS 5, even though it has overall lower melt extent compared to MS 1. Hence, the maps indicate that MS 1 and 5 exhibited the most extensive and longest duration melt events during the SMAP observation period, respectively. The other melt seasons



experienced more typical melt extent and duration (Tedesco, 2009; Tedesco, et al., 2007). The exceptions are an approximate
       10-day melt even over the narrow strip along the Transantarctic mountains to the east of the Ross Ice Shelf during MS 2, and
       the recurring melting on the Amery Ice Shelf.

       In comparison, previous melt detected areas with the number of melt days derived using higher frequency SMMR and SSM/I
       TB retrievals (Munneke, et al., 2012; Picard, et al., 2007; Picard et al., 2006; Anon., 2003) are also shown compared to the
SMAP melt detection results for each season in Fig. 13. The melt areas are in general the same across the continent for both
       frequencies in each melt season. However, the lower frequency of the SMAP L-band radiometer provides additional spatial
       information due to its deeper penetration depth. Because even a small fraction of surface melt will quickly saturate the higher
       frequency signals, they exhibit a larger uniform melt area. For example, the Ross Ice Shelf in MS 1Fig. 15 shows the evolution
       of the melt extent over Antarctica from Oct 31, 2015 till May 31, 2020. While the timing of the maximum melt and the overall
duration of the melt season is fairly consistent from year to year, the curves show how MS 1 has a large spike in the extent
       whereas the curve for MS 5 is broader corresponding to the longer duration of the large-scale melt. The curves also clearly
       show the lower melt extent of MS 3 and MS 4.

       Columns 3 and 4 of Table 5 show the melt area percentage and median of the number of melt days derived from the empirical
       algorithm over Antarctica in each MS, respectively. Fig. 18 illustrates the results in columns 3 and 4 of Table 5 in a bar chart
format. The melt maps show the melt extent and duration across Antarctica, but they do not convey the intensity of the melt,
       which we derived using the snow wetness retrieval and discuss in the next Section.

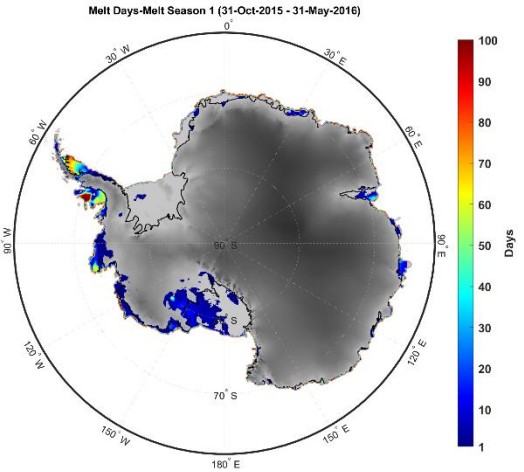

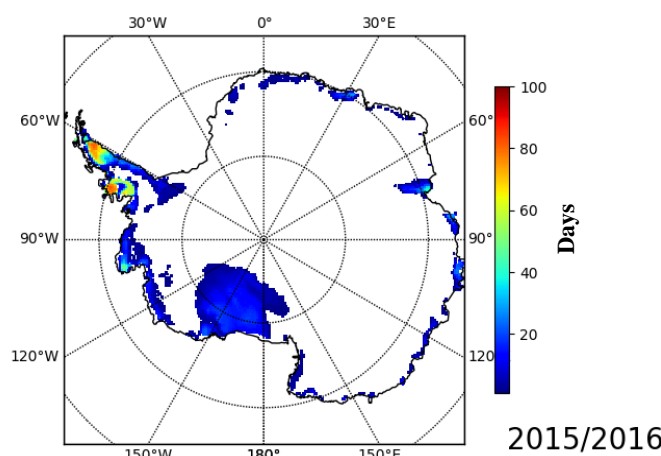

(a)



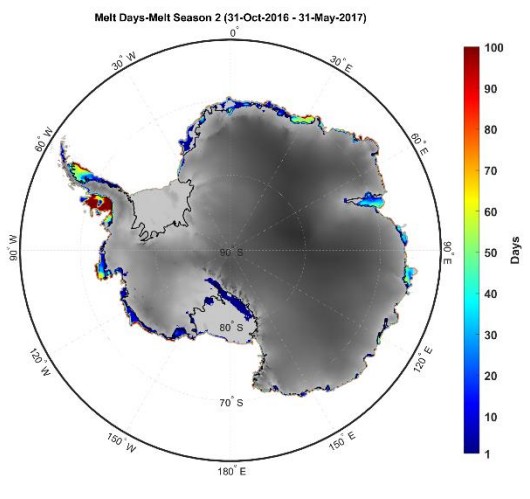
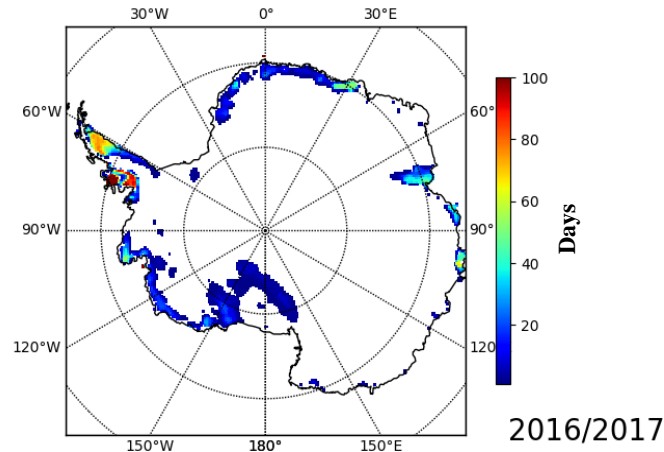

(b)

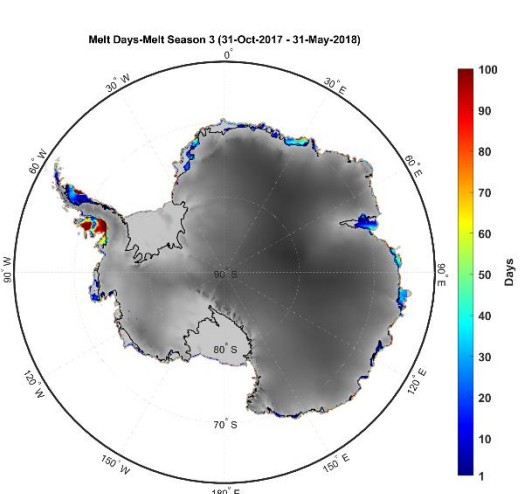
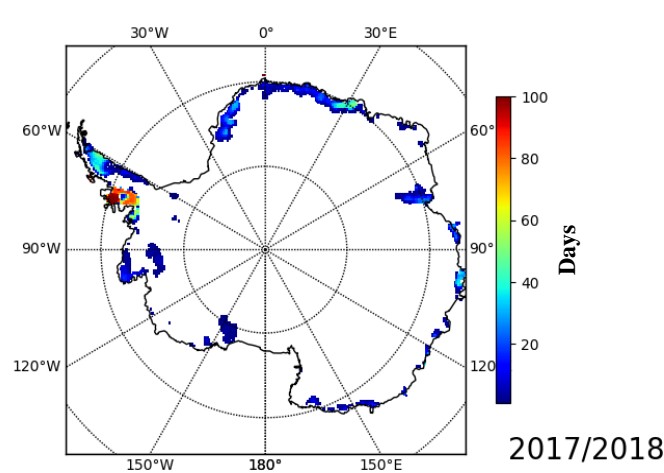

(c)



(d)

(e)

**Fig. 13.** Number of melt days derived from the empirical algorithm using SMAP L-band (1.4 GHz) radiometer TB retrievals (left) and (right) the higher frequencies (19GHz, 37 GHz) of SMMR and SSM/I over Antarctica (http://pp.ige-grenoble.fr/pageperso/picardgh/melting/) for (a) MS 1 (b) MS 2 (c) MS 3 (d) MS 4, and (e) MS 5austral melt seasons between 2015 and 2020.




**Table 5. The total melt area percentage and the median of the number of melt days derived from SMAP L-band TB retrievals using the empirical algorithm and the median of the snow wetness percentage retrieved using the SnoWR algorithm over Antarctica.**

| Melt Season No. | Total Melt Area % | Median of the Number of Melt Days | Median of the Retrieved Snow Wetness % |
|---|---|---|---|
| 1 | 10.17 % | 7 | 0.13 % |
| 2 | 8.80 % | 20 | 0.21 % |
| 3 | 6.39 % | 23 | 0.15 % |
| 4 | 6.33 % | 18 | 0.12 % |
| 5 | 8.47 % | 24 | 0.29 % |

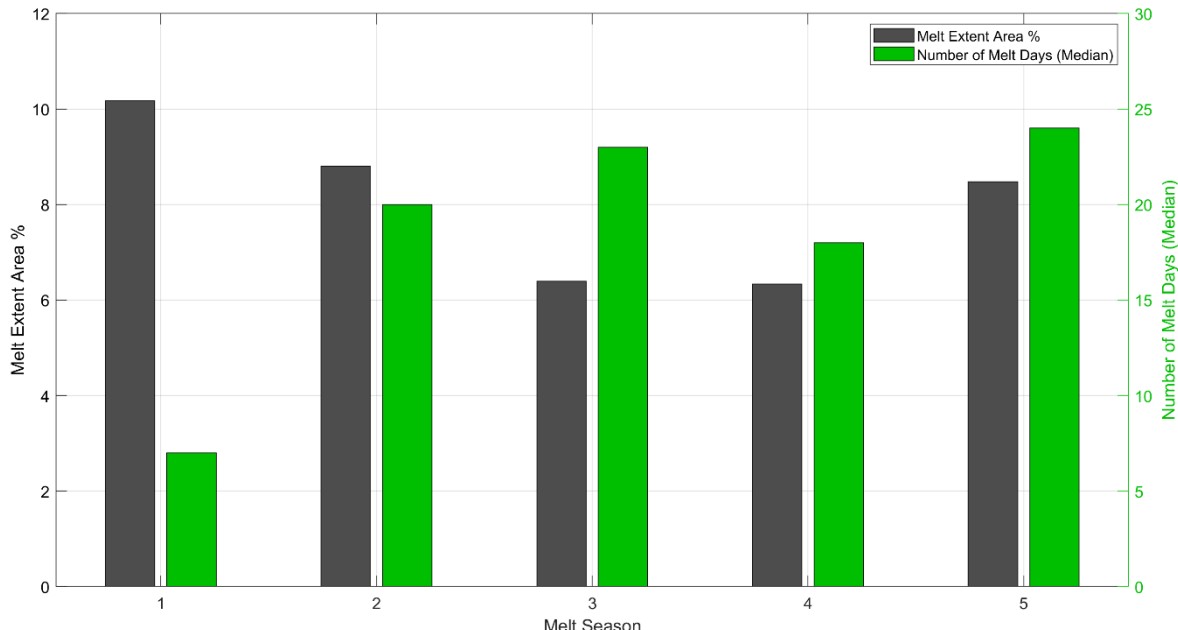

**Fig. 14. Total melt area percentage and the median of the number of melt days derived from the empirical threshold algorithm over Antarctica for 5 austral melt seasons between 2015 and 2020.**

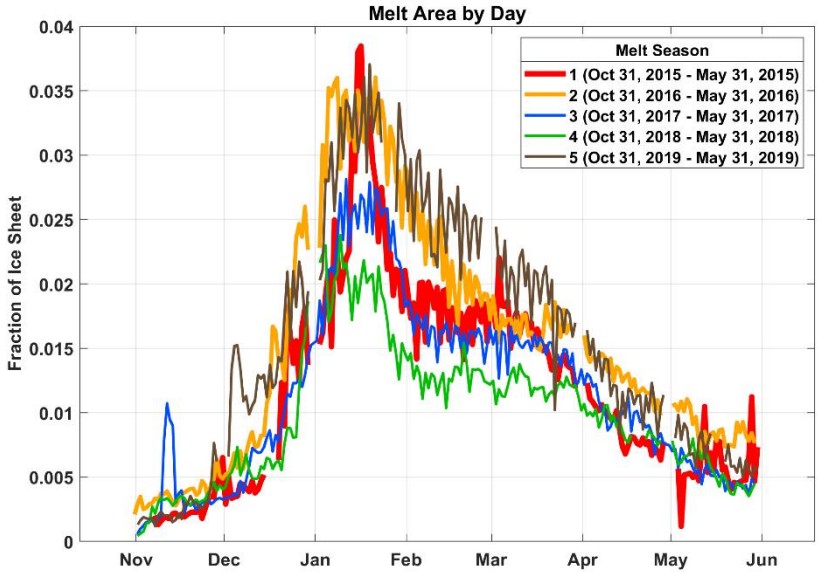

**Fig. 15. The daily melt area derived from the empirical algorithm using SMAP L-band microwave radiometry between 2015 and 2020. The austral melt seasons are color coded.**

### 5.2    Snow Wetness Retrieval Results

Using the SnoWR algorithm, as explained in Sect. 4.2, snow wetness is retrieved over Antarctica for each austral melt season. Only the pixels with significant $\Delta NPR$ and $\Delta T_{BV}$ are processed, as explained in Sect. 4.2, while the rest of the pixels did not show a detectable melt event. The melt pixels in the SnoWR algorithm here are the same as the empirical algorithm results, as described in Sect. 5.1.

Fig. 16 shows the temporal mean snow wetness percentage retrieved across Antarctica. The figure shows that the Ross Ice Shelf melt event was less intense as compared to the melting of the ice shelves along the periphery of the Antarctic Peninsula. In addition, these ice shelves experienced more intensive melt in MS 5 compared to the other melt seasons, as shown in Fig. 17. For example, the Larsen C and Larsen D ice shelves experienced intense melting during MS 5 compared to the other melt seasons. Even though MS 1 exhibited the most extensive melt area, MS 5 had the longest duration and most intensive melt

events. Column 5 of Table 5 shows the median of the snow wetness percentage over Antarctica retrieved using the SnoWR algorithm in each melt season. Fig. 18 illustrates the results in columns 3 and 5 of Table 5 in a bar chart format. The retrieved wetness of MS 5 was clearly anomalous compared to the other melt seasons corresponding to the exceptional melt events in early 2020 (Robinson, et al., 2020). MS 2 and MS 5 have similar melt extent and duration, but the wetness of MS 5 separates it from the more typical seasonal melt of MS 2. In particular, the Antarctic Peninsula and the Amundsen-Bellingshausen Sea

coast experienced very intensive melting, which corresponded to warm temperature anomalies in February 2020 indicated



from reanalysis data (Robinson, et al., 2020). There are other satellite-based studies, such as SSM/I, which shows agreement in terms of snow wetness extent over the Antarctica Peninsula (Zheng, et al., 2019), as well as some in-situ wetness measurements on the east Antarctica in the range of 0 to 4.63% with an average 0f 0.75%. These in-situ measurements were collected during September and October 2007 (Willatt, et al., 2010).


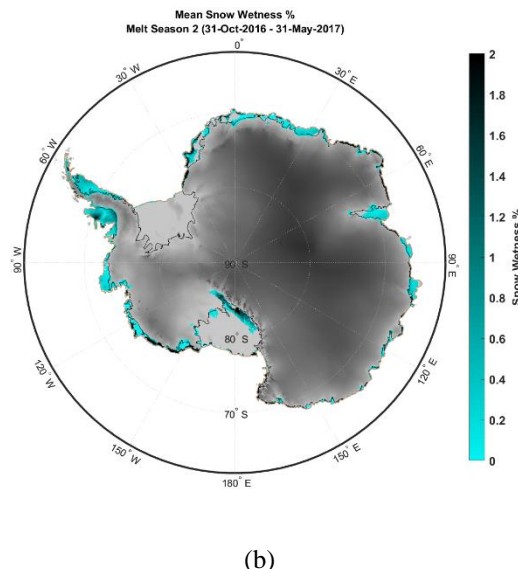

(a)

(b)

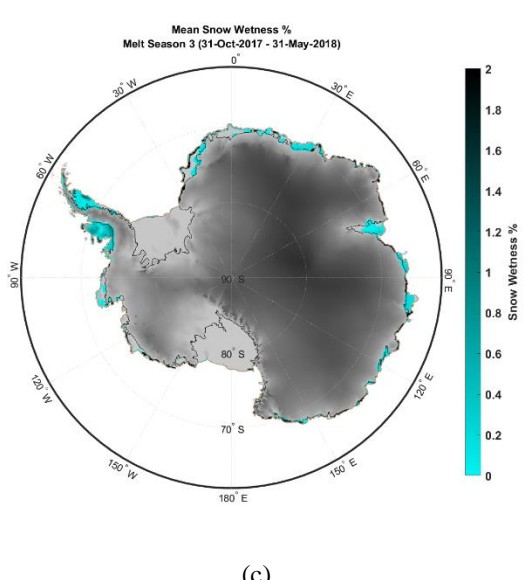

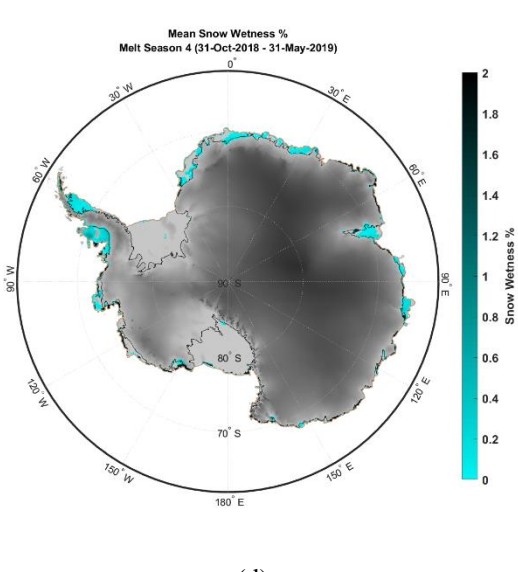

(c)

(d)





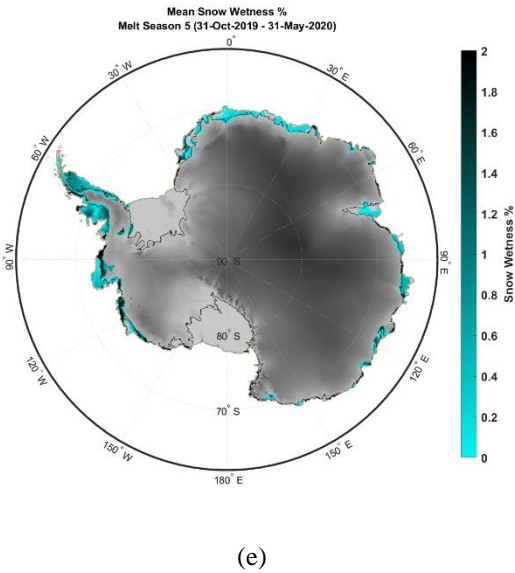

(e)

**Fig. 16. The SMAP retrieved temporal snow wetness percentage using the SnoWR algorithm over Antarctica during the (a) MS 1, (b) MS 2, (c) MS 3, (d) MS 4, and (e) MS 5 austral melt seasons.**

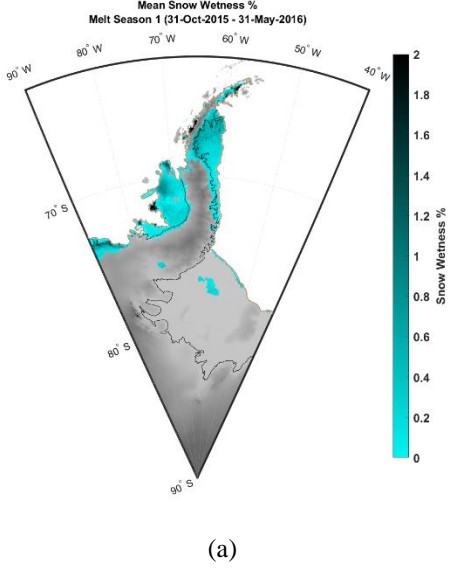

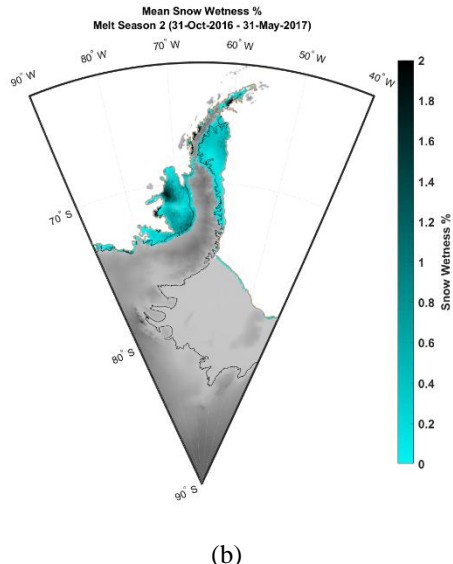

(a)                                              (b)



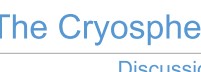

(c)

(d)

(e)

**Fig. 17. The SMAP retrieved temporal snow wetness percentage using the SnoWR algorithm over the Antarctica Peninsula during the (a) MS 1 (b) MS 2 (c) MS 3 (d) MS 4, and (e) MS 5 austral melt seasons.**



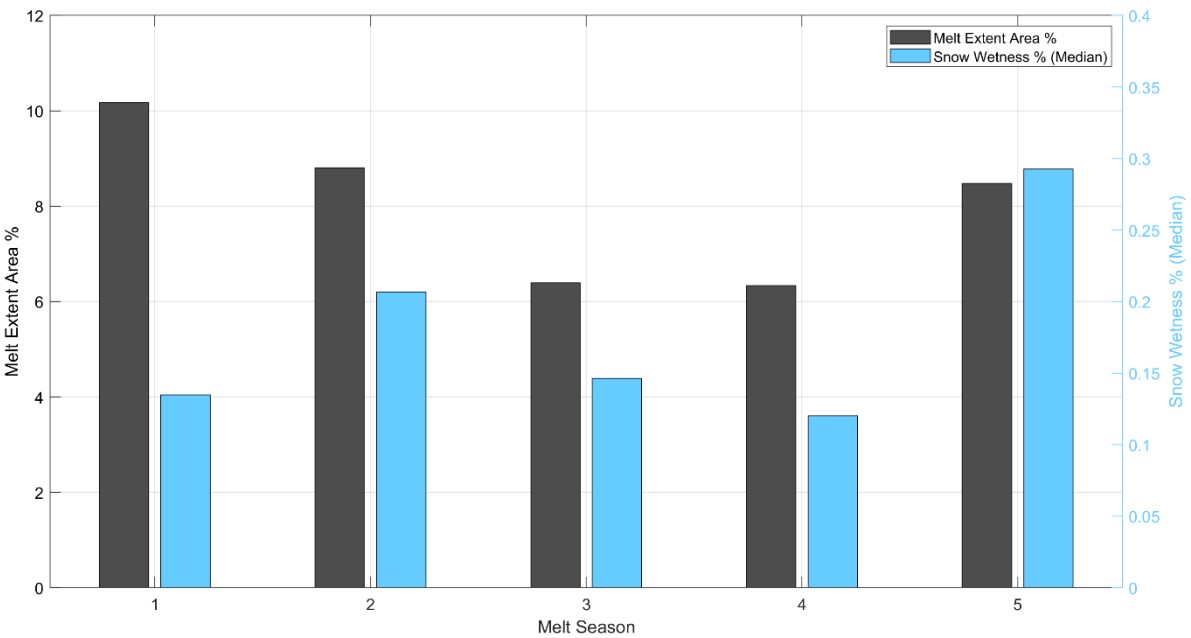

**Fig. 18. Bar chart for the total melt area percentage and the median of the snow wetness percentage over Antarctica retrieved from SMAP TB observations using the SnoWR algorithm.**


## 6    Conclusion

The ability of the SMAP L-band radiometry as a powerful passive microwave remote sensing tool to detect ice sheet melt events was demonstrated. We introduced a new way of computing Antarctica melt extent, duration and intensity using the SMAP L-band brightness temperature observations. The approach exploits the effect of the liquid water in the surface layers

of the ice sheet on the L-band radiation. The long wavelength allows tracking a range of wetness conditions across the surface layers, which provides additional value compared to shorter wavelength observations that saturate quickly with the presence of liquid water in the shallow layers of ice sheet. The results showed that while the extent and duration of the melt during 2019-2020 melt season was not exceptional, the intensity was substantially higher than in other years observed by SMAP, including 2015-2016 melt season which had an exceptionally large melt extent.




## Appendix A


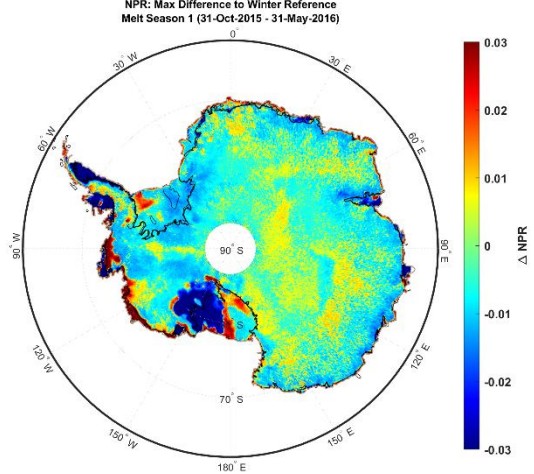 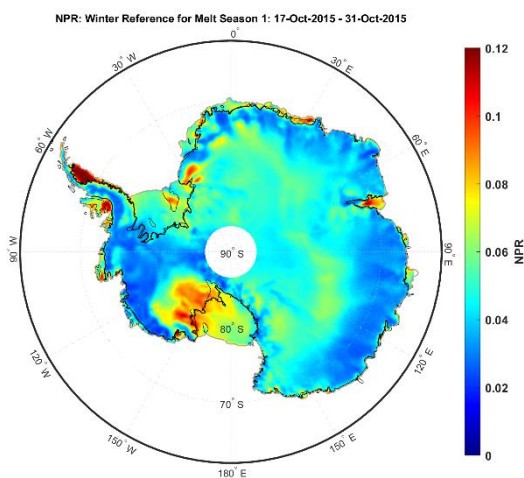

(a)

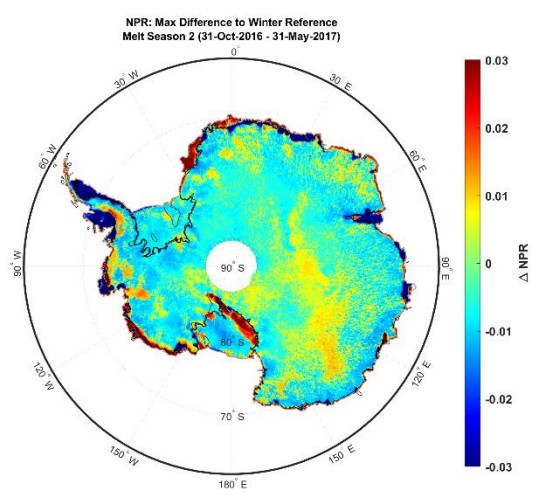 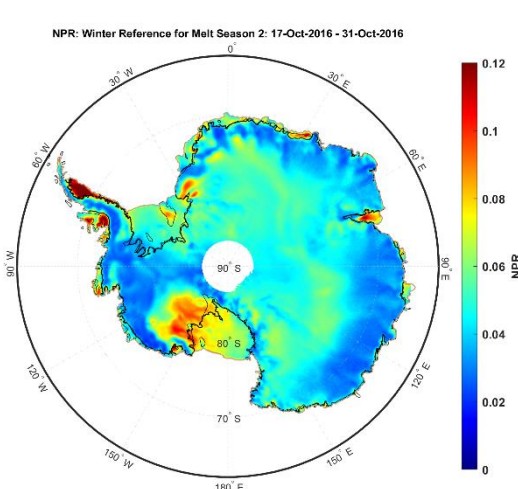

(b)



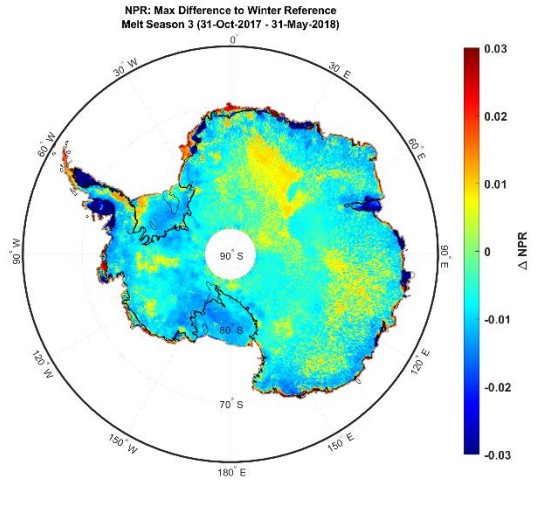
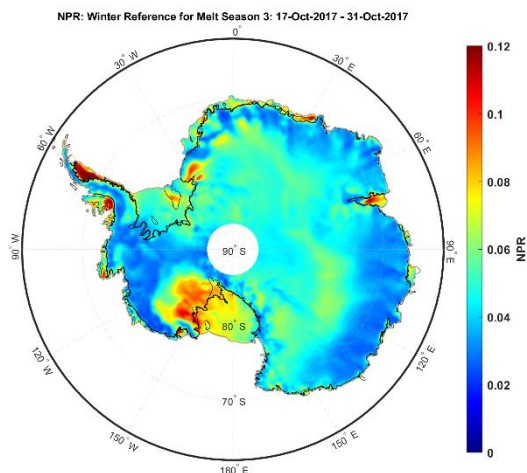

(c)

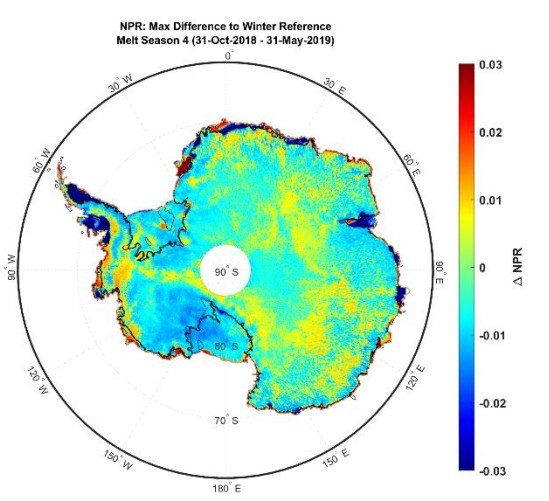
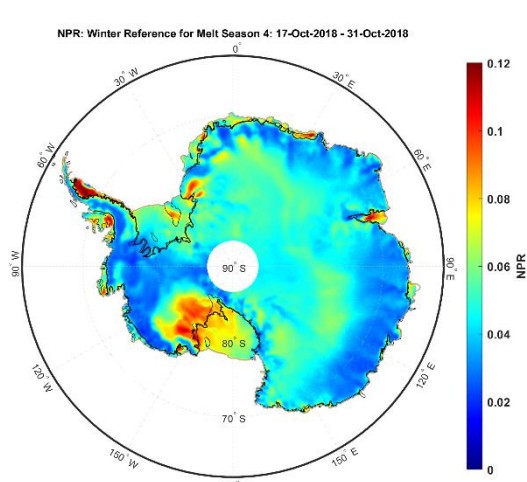

(d)





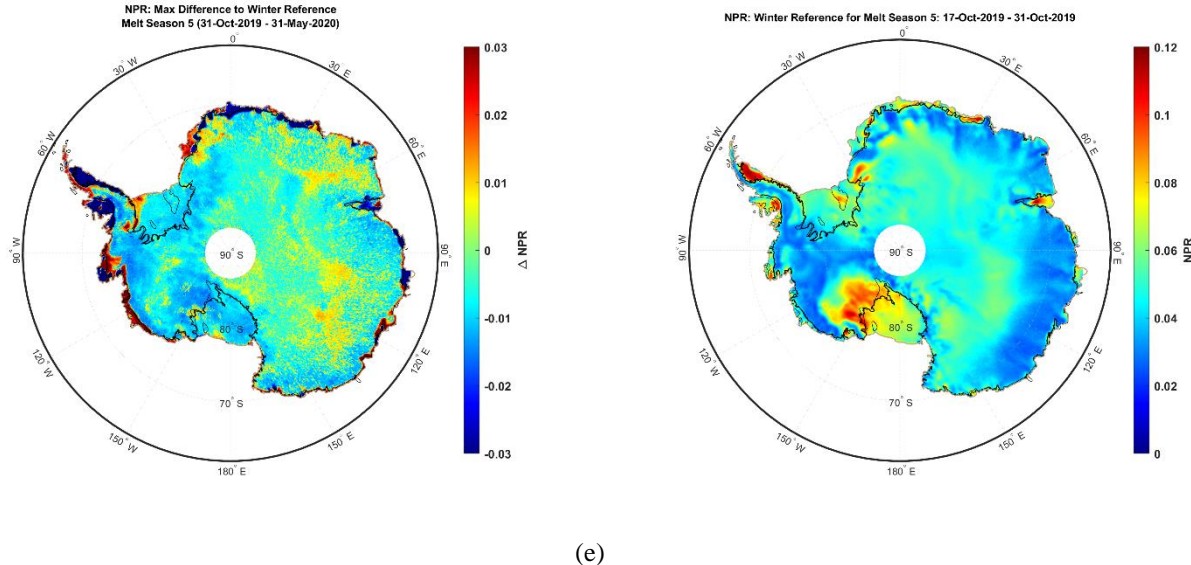

(e)

**Fig. A1 The temporal maximum $\Delta NPR$ (left) and the $NPR_{ref}$ (right) measured by SMAP L-band microwave radiometer over Antarctica for the (a) first, (b) second, (c) third, (d) fourth, and (e) fifth austral melt seasons from 2015 till 2020.**

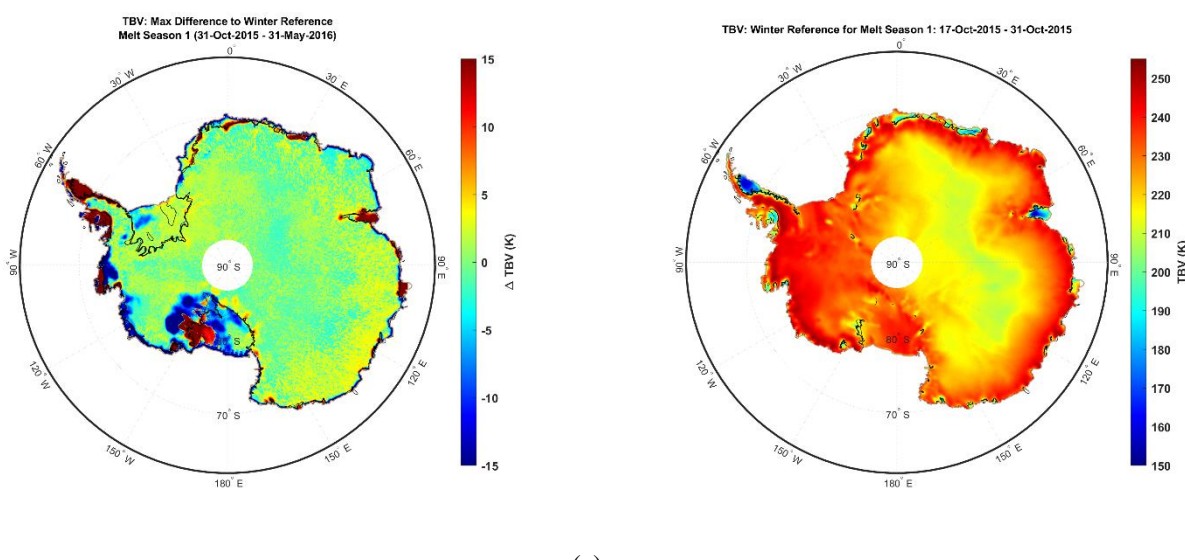

(a)



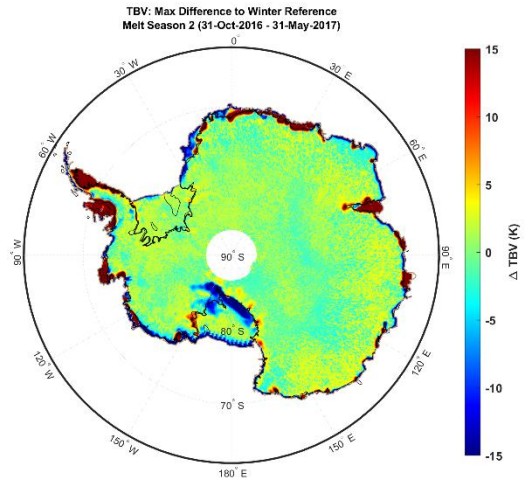
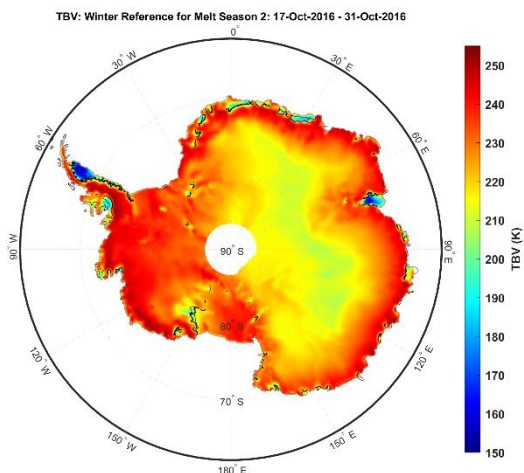

(b)

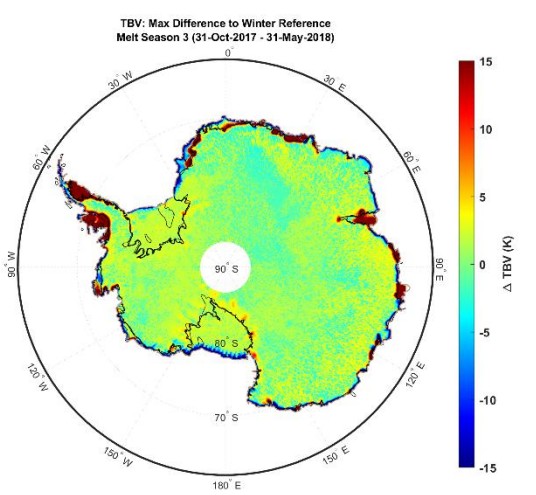
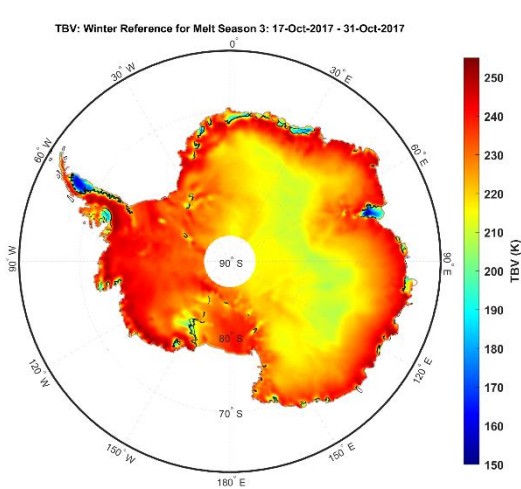

(c)





(d)

(e)

**Fig. A2. The temporal maximum $\Delta T_{BV}$ (left) and the $T_{BV_{ref}}$ (right) measured by SMAP L-band microwave radiometer over Antarctica for the (a) first, (b) second, (c) third, (d) fourth, and (e) fifth austral melt seasons from 2015 till 2020.**

## 7  Acknowledgments

The research described in this publication was carried out at the Jet Propulsion Laboratory, California Institute of Technology, under a contract with the National Aeronautics and Space Administration. This work was also supported by the NASA MEaSUREs (#80NSSC18K0980) and Cryospheric Science Program (#80NSSC18K1055).





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
