# Peer review of "A Novel Approach to Map the Intensity of Surface Melting on the Antarctica Ice Sheet using SMAP L-Band Microwave Radiometry"

_The Cryosphere, 2020_

## Referee Comment (RC1) · Anonymous Referee #1 · 24 Dec 2020

The paper deals with the detection and quantification of snow melting in Antarctica. The topic is very timely given the changes that the continent is facing due to the climate forcing. Basically, the work done is based on two synergic "modules". Half of the work is focused on the identification of the regions experiencing seasonal melting, the onset and the phenomenon and its ending; the other part is focused on the retrieval on the liquid water content of snow. Albeit being synergic, the two parts are self-standing can be discussed separately. Hereinafter general comments will be provided on the two parts, followed by a detailed discussion of the paper. The first part of the paper describes the algorithm used to detect melting areas and the duration in Antarctica. The algorithms leverage previous results well established obtained over both Alpine

(e.g. Macelloni et al. 2005, doi: 10.1109/TGRS.2005.855070.) and polar regions (e.g. Picard et al. 2007 doi:10.3189/172756407782871684). The physical mechanism behind the melting detection is clear and only an extensive validation is lacking, for instance by using air temperature records from AWS or reanalysis. The second part deals with the quantification of the liquid water content in the snow and it is the most problematic. Mainly because the model and its inputs are partially described, and overall any validation is provided. This last point is the most impacting. Even if I recognize that a validation can be difficult, providing just a parameter estimate is not a good practice. This is what stopped many scientists in this effort.

Hereinafter, a point by point review of the paper follows:

Line 44 – The high penetration capability (up to hundreds of meters) is true only for dry snow or the ice of the ice sheets. When the snow contains liquid water or the ice is saline (e.g. sea ice) the penetration depth of e.m. waves at L-band strongly decreases to values on the order of 1m or less. The text must be improved.

Line 73 – Here and in the following: the cited Ulaby & Long book has almost 1000 pages and contains all of the main topics of microwave remote sensing. Citing this book without referring to the specific paragraph or the page is not a good practice. Please add detailed references. Also, some more specific papers can be cited like DOI: 10.1109/TGRS.2005.855070 or doi:10.5194/hessd-9-8105-2012.

Line 105 – This section should be shortened. The mathematical derivation of the final expression can be found in several books and derived easily (e.g. the cited Ulaby & Long, 2014; Tsang et al. "Scattering of Electomagnetic Waves...", 2000). It would be enough to report the final formulae for the brightness temperature. Also, Antarctic snow covers present a characteristic layering and parameters profile (e.g. temperature, density, etc) that must be account in the modelling (e.g. Leduc-Leballeur et al, 2015 DOI: 10.1109/TGRS.2015.2388790, Brogioni et al, 2015 DOI: 10.1109/JS-TARS.2015.2427512). Any approximation that can simplified the modelling must be

justified.

Line 111 – Volume and surface scattering can be neglected for the snowpack that never experienced melting. However, in coastal regions where snow melting happens seasonally, volume scattering can be appreciable because of the metamorphism taking place in the top snow layers.

Line 149 – This section should have addressed the main point of the modelling, i.e. the validation of the model, but it is limited to show that models with two given set of parameters can simulate the NPR behavior. The basic point is that many sets of input parameters exists that can provide a similar behavior. A throughout and rigorous validation of the model has to be provided to substantiate the validity of model estimations.

Lines 156 and 161 – It is not clear the origin of these parameters: if they come from conventional measurements made by the authors or from a publication. Are they representative of the Antarctic coastal regions? Some snow characteristics usually change throughout the melting season, for instance the density of the superficial layers. 450Kg/m3 is typical of aged snow layers that already underwent many melting/refreezing cycles while at the beginning of the melting season superficial snow density remains more or less the same as in the winter (about 250-300 Kg/m3). Why the wet snow layer thickness has been set to 3 cm? The same observations apply the parameters of medium 3. Another fundamental point: which is the permittivity model used? The reference provided doesn't help. In the years several models have been proposed and validated, and their estimates can differ appreciably. Why this one has been chosen?

Line 171 – As the authors show in the paper, melting take place only in the coastal regions and on the ice shelve, which are a limited portion of Antarctica (with respect to the size of the continent). Their geophysical characteristics are different from the inner part of the ice sheet. It is not clear the reason to consider "all Antarctic pixels" in the tuning of the empirical algorithms given that pixels that has never experienced melting

outnumber the other ones. This point should be clarified.

Line 201 – This table is redundant and the information already stated in the text. It can be removed.

Line 207 – These plots are important given they provide a first assessment of the algorithm in Antarctica and should be commented more in depth. For instance, melting is supposed to take place whenever the air temperature is above 0°C however the proposed algorithm missed the melting that took place before mid Dec 2015 and after mid Feb 2016. This is not unexpected given that Tb at L-band is less sensitive to melting with respect to Tb at higher frequencies. These observations have already been showed at some symposium (e.g. DOI: 10.1109/IGARSS.2017.8127587) and further analysis/comparisons should be done here.

Line 215 – I found that a complete legend would makes the figure easier to be read. Also, maximum and minimum temperature should be represented with a different style (maybe lighter with the same color): in the present way they seem three different time-series.

Line 220 – Same comment as line 215.

Line 226 – Which are the specific "realistic ranges"?

Lines 230-231 – This implies that liquid water does not percolate in the lower layers. Is it an assumption to simplify the modelling? Is it possible to substantiate it?

Line 234 – NPR can increase if TbV increases or TbH decreases. So it seems that "In NPR-INCR pixels, NPR and TBV change in positive and negative directions, respectively" should be corrected as well as the sentence that follows.

Line 235 – It is not clear if the retrieval algorithm is run for each pixel flagged as melting area, or in a different way.

Line 236 – Which kind of thickness is retrieved? Thickness of medium 1 and 2?

Line 239 – ". . .,by comparing. . ."which type of comparison has been made? Has a cost function been used? In case, which is its formula?

Line 240 – ". . .there is no vertical layer structure during 240 the FS in the dry snow layers. . ." this is true only for the very simple model used. Actually a real snowpack has many layers that must be considered in an accurate modelling of the scene (Leduc-Leballeur et al, 2015 DOI: 10.1109/TGRS.2015.2388790 and Brogioni et al, 2015 DOI: 10.1109/JSTARS.2015.2427512). If the layering is disregarded it is impossible to re-produce the microwave angular signature of the dry snow emission. This was a prob-lem highlighted by the SMOS team at the beginning of the mission. Given the authors use a very simple model, the parameters that they will retrieve can be considered "equivalent" and useful only to get the final product.

Line 242 – In this minimization there are several free parameters and few measure-ments. The problem is clearly ill-posed. How the authors can be sure that the geo-physical parameters found by the minimization are the best set for the snow covers instead of others? The fact that it is possible to simulate correctly the TbV timeseries does not imply that the geophysical parameters found are correct and could provide the subsequent reliable liquid water content estimation (actually the fitting of TbH shown is poor).

Line 246 – Why in the fitting process the authors consider H and V pol SMAP timeseries independently instead of searching for a inputs that fit both of them together? Given a set of ground parameters the model should be able to simulate both polarizations. If not the case, the model is not accurate enough, the input parameters are incorrect, or both of them.

Line 251 - It is somewhat weird that the ERR() operator stands for a temporal mean. Is the text correct?

Line 252 – It is not clear between which quantities the bias can be found. Could the authors comment this sentence?

Line 252 – It is not clear the aim of this operation. The liquid water content, along with the other inputs of the models, has been derived by fitting the SMAP measurements. Then the same retrieved geophysical parameters are used to simulate the Tb with the same analytical model and assess in this way the reliability of the method. If all the input parameters come from the fitting and the analytical model is the same, why the Tb simulated should not agree with SMAP measurements?

Line 256 – what is mv2ref?

Lines 268-270 – This seems to be more related to my previous comments than to the validity of the method. The plot seems to suggest that SMAP TbV timeseries is used to determine the input parameters of the model, that in turn produces the best Tb match.

Lines 271-275 – Here lies the other major weakness of the paper: there isn't a validation of the algorithm. The results are plausible but a validation has not been performed. The fact that the liquid water content of the snowpack ranges between 0 and 2.5% is encouraging, but it is more or less the same range of common alpine snow. Also, the referred paper deals with snow ontop of sea ice which undergoes to different processes with respect to snow on land. I am well aware that validating a retrieval method in Antarctica can be a tough task but it is to be done.

Line 296 – Table 4 is useless: it contains the same timeframe for all the seasons. The information can be stated in the text.

Line 232 – As said before, it is not only matter of higher penetration depth. Indeed the timeseries represented shown that L-band starts detecting the melting later than the physical onset (air temperature above 0°C) and stops earlier. It is more a different interaction between e.m. waves at these wavelengths and the snow/ice medium.

Line 373 – Willat et al. (2010) performed measurements on the sea ice off the East Antarctica coast and not on land. As said previously, the LWC range of 0-5% is characteristic of the snow in general: above few percent of LWC liquid water start to percolate

in snowpack so it is difficult to find higher values.

Summarizing my review, I found that the paper is composed by two parts being the mapping of regions experiencing melting more advanced that the liquid water content estimation. This latter part is at an early stage, lacks of a proper validation and it is not ready to be published. Even if all of my comments will be addressed (either because I misunderstand the paper or the authors improve it) it is not possible to publish a method that is not validated. This is the weakest point that must be addressed properly.

---

## Referee Comment (RC2) · Anonymous Referee #2 · 12 Jan 2021

Review of

A novel approach to map the intensity of surface melting in the Antarctica Ice Sheet using SMAP L-Band microwave radiometry

by

Mousavi, S., et al.

Summary: Satellite passive microwave observations have since long been used to map surface melt on ice sheets. Such observations are independent of daylight, widely insensitive to atmospheric influences and provide sufficient spatio-temporal coverage.

[Figure]

This contribution deals with such observations carried out at L-Band frequencies of the SMAP satellite, available for Austral melt seasons 2015/16 through 2019/20. A novel empirical approach is developed to map the area affected by melt employing the brightness temperature polarization difference and the vertically polarized brightness temperature observed from SMAP. Obtained maps of the melt areas are shown together with those of another existing product. Another empirical approach is developed, based on two simple layered models, to estimate the snow wetness resulting from the surface melt. Maps and some simple statistics of the snow wetness estimates are shown.

General Comments (GC): GC1: While this contribution contains interesting elements of how to use SMAP L-Band brightness temperature measurements to obtain an estimated of surface melt area and associated snow wetness, its added value for the scientific community is not yet worked out adequately. One of the main advantages of an L-Band sensor is its larger penetration depth into the snow / firn / ice system of Antarctic ice sheets, ice shelves and glaciers. This is where using L-Band could provide additional information over using brightness temperatures obtained at the classical microwave frequencies used for such melt parameter retrieval, i.e. near 20 GHz and near 37 GHz. Because this contribution does not adequately introduce and discuss the interactions between the physical properties of the snow / firn / ice system that are relevant for its microwave remote sensing - especially the differences that can be expected from utilizing the different frequencies (around 1 GHz versus near 20 / 37 GHz), this contribution cannot convince with credible conclusions about advantages or disadvantages of using SMAP data for the proposed purpose.

GC2: The description of the two approaches appears not to be based on a solid foundation of the relevant physics, is not easy to follow and should be illustrated more by examples of intermediate retrieval steps; it should also be better tied to the shown sample maps of NPR and TB_v-pol. See my numerous specific comments regarding this issue.

GC3: The interpretation and discussion of the results appears to be rather rudimentary. The results are not overly well backed up with comparable data or additional data of, for example, atmospheric conditions. Also the presentation of the results should be improved. Also here I have numerous specific comments.

GC4: This contribution lacks a critical discussion of the developed approaches and their results (pitfalls, limitations, applicability ranges), an evaluation of the results - especially with respect snow wetness. The discussion lacks consideration of influence of topography on SMAP observations. Finally, the added value of how and where such L-Band measurements provide additional high-quality information to the existing passive microwave products - e.g. because of being less sensitive to first signs of surface melt but more sensitive to melt water deeper in the snow / firn system is not appropriately demonstrated.

GC5: The reference list needs to be revised as many elements are missing.

Specific Comments: Lines 17/18: While I agree on this statement I am wondering whether this signal saturation is an issue for the relatively cold conditions on the Antarctic Ice Sheet. One could ask whether this is really snow melt or whether it is snow metamorphism. In addition, how thick is the snow cover on the Antarctic ice sheet that penetration issues are important?

Lines 20/21: Rephrase. Not clear whether these are TBh-differences at different times or TBv minus TBh differences at one time.

Line 35: I suggest to add a reference that the uncertainty of the contribution of Antarctic ice sheet / ice shelf melt to sea level rise is particularly large, meaning that any improvement in the understanding of melt processes would be of advantage. At the same time I suggest to make a clear statement that the majority of Antarctica's ice loss comes from glacier / ice shelf calving and basal melt of glaciers and ice shelves rather than from surface melt. It is important to make clear that studying surface melt is important to better understand precursors for ice shelf disintegration but that - in contrast

to terrestrial glaciers and Greenland - it is not that relevant for SMB considerations.

Line 50: "may warm to depths of about 3 meters" –> This is a little unspecific. Warming to which temperature? Where can and does such warming occur in terms of the geographic region and altitude?

Line 66: "are gridded ..." –> Is this the EASE2.0 grid? What is the sampling distance along and across track? This information is required to provide credibility to the comparably small grid cell size of the used grid. In addition: Is this gridded TB product a daily average or are data of ascending and descending overpasses gridded separately such that there are two maps of TBs every day? Please add the respective information to your manuscript.

Line 67: "orbit" –> I suggest to add the orbital inclination to underline that SMAP has a near polar orbit but that there is a disc without observations centred at the pole.

Line 69: "The radiometric resolution ..." –> Does this refer to the 9km product? If so, is Piepmeier et al the correct reference then?

Lines 79/80: "it does not ..." –> I agree with this statement as far as it is based on the fact that TB is the product of physical temperature and emissivity. However, what happens if the relevant physical temperatures are different for the h- and v-polarized channels? The penetration depth into the snow / firn / ice may depend on the polarization. With that the main contribution to the actually measured TB can originate from different depths at H-polarization than at V-polarization. While this difference is possibly negligible at near-90 GHz frequencies, it starts to play a role at near-20 GHz frequencies and likely even more so near 1 GHz. Presumably this applies in particular to dry and cold conditions and the difference is likely to vanish once the snow is wet, but I am sure that it creates noise in the transition seasons, i.e. those periods which are critical to accurately determine onset of melt and freeze-up. Please comment on this issue in your manuscript.

Line 82: "maximum NPR seasonal difference" –> Please provide a physically sound motivation about why you use / show the maximum NPR difference.

Line 83: "Oct 31 ... May 31 ..." –> Why not Nov 1? What is the motivation to expand the melt season into winter? Wouldn't March 31 have been more than sufficient given the fact that sunset is March 21 and freezing commences in February / March?

Line 84: "Oct 17 - Oct 31" –> Please explain why you used a time period after sunrise to determine the reference NPR value. Wouldn't it be more accurate to use a period of the winter months, i.e. July or August?

Lines 88/89: "cause the NPR to increase ... conditions." –> What is "V and H polarization difference"? Do you mean NPR? What is the physical mechanism causing a larger NPR during intermittent melt than during cold & dry winter conditions? Wetness is expected to increase the emissivity at both polarizations. For an increase of the NPR the V-pol emissivitiy needs to increase more than the H-pol emissivity - which is rather unlikely, isn't it? Please be more specific with this explanation and provide a reference which clearly supports your notion.

Lines 90/91: "complex subsurface structure ..." –> This reads as if the penetration depth is only reduced to tens of meters in areas with these complex subsurface structures - which is I doubt is the case because melt conditions generally decrease penetration depths no matter what is underneath.

Figure 1: I suggest to swap panel (a) with panel (b), i.e. show the reference left and the difference right. I suggest to use a non-rainbow color table in the reference map and a blue-white-red color table in the difference map. I note that the difference maps seem to have quite some values outside the value range displayed by the legend. Given the fact that the pattern of the differences is quite noisy for the value range -0.005 to 0.005 you would possibly not loose too much information if you'd expand the value range of Delta NPR to -0.05 to 0.05. I suggest to overlay isolines of the altitude onto the reference map. See my comments in the text with respect to the choice of time periods. Please

add to the caption an explanation for the white area centred at the pole. Please add to the caption information about the coastlines / ice shelf termini shown. Please also refer to the appendix where you show this map again plus maps of the other melt seasons considered.

Section 3: Overall, I find the description of the physics involved not convincing. What is missing is an overview about the physical properties of the snow / firn / ice system determining the radiometric response.

How thick is the snow / the firn layer? Certainly it varies, sure, but how much and what is the influence on the L-Band measurement and your retrieval?

What is the vertical density gradient in the snow / firn layer? As permittivity is a function of wetness AND density this parameters needs to be taken into account. In this context I suggest to provide details about the permittivities used instead of just referring to Ulaby & Long, 2014.

What about the radiation emitted from the ice underneath? Currently, the deepest layer is a "dry snow" layer while the actual radiation might emanate from the ice body of the ice sheet or shelf.

Please provide a reference that surface scattering and volume scattering can be neglected. I am missing inclusion of the cosmic background radiation which enters / is reflected somewhere in the snow/firn/ice system. While this term can be neglected at higher frequencies it cannot at L-Band.

Despite substantial topographic complexity your are assuming horizontal interfaces at the scale of the footprint (about 35 km x 35 km). While this assumption is valid on the plateau and on the ice shelves it is not valid elsewhere. In addition, in topographically complex terrain the interaction between topography and the varying local incidence angle can possibly not be neglected but is not discussed or even mentioned in this section.

Lines 139/140: I suggest to provide more detail here. Why can these two layers be combined? How much of the radiation from layer 4 is actually entering layer 2 when layer 3 is a highly absorptive medium? What physical parameters does this layer 3 have in comparison to the wet-snow layer 2 that it can be termed a highly absorptive layer? Wouldn't a wet snow layer be exactly such a layer? The description of this concept is, in my eyes, not sufficiently well backed up by actual physical snow / firn layer properties.

Again, T_0eff should better be T_eff; furthermore I would avoid to use "eff" as a subscript to "p" in the reflectivity and instead of Gamma_p_eff use Gamma_p,eff

Section 3.3: I have difficulties to adequately understand the motivation behind choosing these examples. Please be more specific and try to tie these examples as close as possible to measurements on the Antarctic ice sheet / ice shelves. It is not clear why you choose for the 3-layer model a 3 cm thin snow layer while for the 4-layer model a 25 cm thick snow layer is chosen. It is also not clear why the high absorptive layer has a thickness of 50 cm.

The dielectric constant given in Table 2 for layer 3 needs to be better motivated. Ideally you do this by explaining that (wet) snow is a mixture of pure ice and air (and liquid water), provide values for the real and imaginary parts of the pure materials and adequately derive and explain the value chosen. In the same context, it would make a lot of sense to see what the dielectric constants used for the other layer have been. In general it would help the reader a lot if you'd have provided information upfront about typical penetration depths of microwave radiation at L-Band frequencies into dry and wet snow.

Figure 3: Please explain and discuss by means of the inherent physics why the brightness temperature at L-Band decreases with increasing snow wetness. This is contrary to what is observed at other microwave frequencies, say near 7 GHz or near 20 GHz. I seriously doubt that the model is doing the right thing here.

I suggest, in addition, to relate the results shown in this figure and shown in Fig. 4 to specific regions in Fig. 1 (and the respective figures in the Appendix). This would at least help to understand that the 4-layer seems to be applied best to ice shelves whereas best examples for the 3-layer models are possibly found somewhere along the steep topography.

The unit of the TB is "Kelvin". Please correct in this as well as in the other respective figures.

Section 4.1: I can understand that you want to use your own algorithm - even though it appears to be way more complicated than the more simple algorithms developed to detect, e.g., melt onset, duration and freeze-up on Arctic or Antarctic sea ice or the pan-Arctic in general (e.g. Willmes et al., Arndt et al., Stroeve et al., Markus et al., Wang et al.), fine. But what one can expect is, that you discuss existing algorithms for such a purpose, trying to delineate why your method has an advantage. Currently, I have difficulties to see this advantage. What is especially strange to see is that you are working with a pan-Antarctic spatial average in Eqs. 12 and 13.

I suggest to stress that it is important to use the absolute value of the differences in Eqs. 12 and 13 to cover both conditions - as shown in Fig. 1 and covered by your two models.

Lines 172/173: Hence the Z-parameters are computed for every grid cell and every summer season?

Line 181: Provide more detail about this error function, please, not just a citation.

Should I choose a low or high Z-value?

Equation (15): I don't get this concept; shouldn't this be the sum over n_ms values of 1-FAR_1day divided by the square root of n_ms?

Line 185: The notion of the period reminds me again that you did not yet adequately motivate why you did not start with Nov. 1 and why you extent the melt season way into

the winter. Figures 6 and 7 are very good examples to illustrate that replacing May 31 with March 31 appears to be a viable solution.

Lines 186/187: You want to have a small FAR and therefor you choose a Z which guarantees this, i.e. a large Z. I don't think that you assumed things here.

Line 190: Why is it important to find a minimum number of days for the winter reference values? Why don't you simply choose winter reference values from typical winter months, e.g. July and August?

Lines 196/197: "Even though ... missing days" –> Why is this? This is not clear.

Lines 197-200: "Our proposed two week ... to 2020" –> Not clear why and how you end up here.

Further up you gave the impression that it'd make sense to estimate the Z-parameters for every grid cell. Here we learn, that these are constant values throughout your study ... 5 and 10 ... hmmm ... ok.

Then you compute a range for DELTA_NPR and DELTA TB_v-pol which is valid for every (?) grid cell for all melt seasons considered ... and these cover a surprizingly small range of 0.001 for DELTA_NPR and 0.34 K for DELTA TB_v-pol. So, in essence you are claiming that - independent of the melt season and, more surprisingly, independent of the geographic region (in the middle of the ice sheet, at a steep topography region, and on an ice shelf) you can always adequately retrieve melt onset with such constant thresholds. Isn't this a surprizing result? Isn't it surprising that a change in NPR by more than 0.01 and of TB_v-pol by more than about 7 K is indicating melt no matter where you are, no matter whether winter TB_v-pol is 150K or 250K or winter NPR is low or high.

Lines 205-207: This needs to be explained. To which snow/ice vertical structure changes is h-pol TB more sensitive than v-pol TB? Could it be that topography plays a role and that the plane layer assumption made is not appropriate in topographically

complex regions? Could it be that snow metamorphism, grain size changes, snow and firn density changes, percolation of melt water, formation of surface crusts and buried ice lenses and layers have an impact here? Please be more specific in your manuscript.

Lines 212/213: Naively one would think that one needs temperatures above freezing, i.e. only the range 0 to +5degC would apply here. Why do you think it is important to write about a range from -5degC to +5degC?

Fig. 6: I suggest to make clear that the cyan bar denotes the typical range within which at that particular location the NPR varies during winter. I suggest to add vertical lines that indicate melt-onset and commence of freeze-up and annotate these with the respective date. The same should be done in Fig. 7.

Line 226: "specific realistic ranges" –> Please provide these specific ranges and the stepsize with which you change which parameter.

Lines 226-231: Please provide more specific physical background here. Why do you perform snow wetness retrieval for frozen cases? Isn't snow wetness essentially zero in these cases? What is the "secret" of the high-absorptive layer in the 3-layer model; it is still not clear what kind of a layer this is in terms of the physical properties.

Lines 253-238: You use the LUT and the observed L-Band TB values to derive both, layer thickness AND the complex dielectric constant? How can you derive the latter (real and imaginary part) with unknown density and liquid water content?

Line 241: "in the dry snow layers" –> ... so infact we do have several snow layers of unknown thickness and density ...

Lines 242-244: Without more physical information about the typical values involved and an illustrative figure documenting which intermediate stages the retrieval runs through, this is not understandable. It is not clear how the retrieval can end up with one unique maximum snow wetness percentage.
Line 245: "another LUT is made" –> Which parameters does this LUT contain? And what are the used values for snow density and temperature as well as the dielectric constants (real and imaginary)?

Line 247: "daily" –> Why daily? Where comes the daily information in? Perhaps you need to be more specific. Why do you "simulate" daily TBs? Apparently the LUT contains a set of articifial TB_h and TB_v values for every snow wetness value (and other unknown parameters' values). For each grid cell you take the time series of SMAP TB_h and TB_v values and take that snow wetness value from the LUT which corresponds to the LUT TB - SMAP TB data pair with minimum absolute difference. Hence, you obtain an estimate of the snow wetness for each day of an unknown time period for every grid cell with a SMAP TB measurement?

Lines 252-254: I don't understand what you do ... you model TB values but you retrieve daily snow wetness ...

Lines 269-270: "The simulated ... see (18)." –> I doubt that this unique conclusion can be drawn without more information about snow and firn density and temperature and dielectric constant. I am not convinced a 1-to-1 relationship exists here.

Lines 274/275: The Willatt et al paper reports upon conditions in sea ice. These might not be representative for snow conditions on an ice shelf or the interior Antarctic ice sheet. You should seek for and find more adequate references.

Line 276: Why is Z_m_v set to 2?

Fig. 11: How thick is this wet snow layer? In the Antarctic, melt water often percolates downwards into deeper snow / firn layers, refreezing there. How sure can you be that the used SMAP TB data observe this site at a time of the day where liquid water is in fact present near the snow surface and/or within the snow cover?

Table 4: This table can be deleted because winter reference and melt season periods cover the same days in all years and have been introduced earlier already. Instead of

MS 1, MS 2 and so on, you could simply refer to 2015/16, 2016/17 and so on.

Line 308: Why "consistent"?

Line 317: Why is a "recurring melting" exceptional?

Lines 318-320: The melt maps shown in Fig. 13, right column are certainly not based on SMMR and SSM/I data. They use the method which is derived based on these satellites. Hence I suggested to correct this sentence accordingly and make clear that this is based on SSMIS data and provide the URL you used in the Figure 13 caption also in the text.

Lines 320/321: "across the continent" is perhaps not the best description. I would say that SMAP and SSMIS melt areas and melt durations agree well - mostly over the ice shelves. "both frequencies" –> I am not sure what you mean by this; Fig 13 right column shows SSMIS results which appear to be based on 19 and 37 GHz data while your results are based on L-Band data.

What you should mention is that the SSMIS data appear to have a coarser spatial resolution - which is clear given the fact that the footprint size is similar (37 GHz) and larger (19GHz) than the one of the SMAP data and the resolution of the grid used is possibly 25 km.

What you could mention is that this might be the reason why the SSMIS data show melt areas that extend more onto the glaciers (e.g. on the Antarctic Peninsula) than the SMAP data do ... albeit this could be an artifact of the SMAP melt area detection as well.

Line 322: "Because even a small fraction ..." –> This is a too global statement and needs more specific information taking into account temperature, snow depth, typical penetrations depths of the used frequencies in dry and wet snow, and the like.

Figure 14: I guess I have difficulties to understand Fig. 14 in the context of Table 5 and Fig. 15. One problem I have is the "Melt extent area" being 10% for melt season

2015/16 in Fig. 14 and Table 5 but peaking at a fraction of the Antarctic ice sheet of 0.035 in Fig. 15 for the same melt season; I would translate a fraction of 0.035 into 3.5% ... not 10%. The second problem I have is to understand how the "median number of melt days" shown in Fig. 14 relates to the much longer periods of melt shown in Fig. 15. I guess I misunderstood the reference region(s) for which these quantities are derived. Perhaps you could describe things a bit better here.

Figure 15: This is one of the key figures of this paper and should be discussed adequately in the context of Figs. 6, 7, 11, and 13. This figure suggests that a considerable fraction of the grid cells considered exhibit a melt duration of 130 days or more. This is not supported by the other figures mentioned.

What is the surface temperature of the ice shelves for which possibly such prolonged melt durations are retrieved by your method? Do the ice shelf surface temperatures in February/March still support melt conditions? Since new (sea) ice formation commences around Antarctica around end of February / beginning of March it seems very unlikely that ice shelves still experience melt conditions.

What explains the "plateau" of elevated melt activity for basically all melt seasons in February / March?

I suggest to, in addition to Figure 13 show maps of the melt onset and commence of freeze-up and comare these with (surface) air temperatures, e.g. from ERA5.

Line 372-374: "as well as some ..." –> As these measurements were carried out on sea ice and at a completely different time of the seasonal cycle I recommend to delete this part of the sentence.

Figure 16: This (in Fig. 16 and Fig 17) is the mean snow wetness over which period? I assume these values are computed only for the period where your algorithm indicated surface melt? Hence, if the melt extended over just 1-2 days it is the mean over the snow wetness values of these days whereas if melt extented over a period from Dec.

10 through March 20, then it is the mean over these about 100 days. Or did you perhaps use a cut-off minimum number of melt days to compute these mean values (and also the median values shown in Table 5 and Fig. 18)?

I note that there are quite sharp gradients in the snow wetness values shown - particularly in Fig. 16. What causes these?

I further notice that the color bar does neither resolve well higher snow wetness values (i.e. above about 1.6) nor is it clear whether the values are much larger in the blackish areas in the maps. Please check whether you could use an more adequate color bar and coding.

Fig. 18: What is your motivation to show the median snow wetness here but the mean snow wetness in Fig. 16 and 17?

Line 385: A critical discussion of the results is completely missing. Neither do the authors comment upon the potential limitations of their method nor do they put their results adequately into context with other parameters. While the comparison between melt days shown in Fig. 13 is convincing, the discussion of the snow wetness results is not. It is not adequately discussed in relation to the spatial distribution of the melt duration. For instance, do melt duration distributions on the Larsen C and B ice shelves show a clear south-north gradient in MS1, MS2 and MS5, which is not reflected in the snow wetness data. The question is why? Gradients in the snow wetness, e.g. on the Bellingshausen / Amundsen Sea ice shelves with near zero wetness close to the seaward edge of the ice shelf but wetness close to or larger than 2% directly along the coast are not discussed. To my opinion, the results call for much more dedication to the relevant physical processes in the snow/firn/ice system and their relation to the L-Band brightness temperature observed by SMAP. My main argument here is that snow melt on ice in the Antarctic is possibly very much dependent on the temperature profile, the snow depth and the physical conditions of the old snow / firn underneath the current snow layer; snow metamorphism, melt-refreeze cycles, meltwater percolation, formation of superimposed ice and ice lenses, topographic compexity and the like need to be taken into account. This would also require an in depth analysis of the meteorological conditions (time series of air-temperature, humidity, wind, precipitation) over the course of the melt season. A wet snow signal could also simply be the result of a heavy summer wet snow snowfall event and might have nothing to do with melting.

In short, while I am ok with the fact that one can detect melt on Antarctic ice shelves and ice sheets using SMAP to some extent (even though also these results need to be discussed critically much more than is currently done), I am not convinced that the snow wetness retrieval provides meaningful results over all regions shown in this paper without a more appropriate consideration of the inherent limitations of L-Band remote sensing of a medium having an unknown penetration depth.

Typos / editoral remarks: Line 27: "are increasing" –> Perhaps better: "have been increasing" and looking for a more recent reference.

Line 37: "n.d." ?

Line 39: "several satellite sensors" –> I suggest to add the term "conventional" in some way here to have a reference point for the statement in Line 52.

Line 79++ / Equation (1): TBv –> Please find one common abbreviation or symbol for brightness temperatures. I recommend to NOT use B as a sub-script. One solution could be to use TB_v and TB_h where v and h are sub-scripts denoting vertical and horizontal polarization, respectively.

Line 87: "Melt" –> "melt"

Line 89: "negative NPR change" –> Better: "decrease in NPR"

Line 94: "After detecting the melt areas ..." –> Please refer to the section of the paper where you describe in detail how you detect melt areas.

Equation (3) ++: Following up with my comment to Eq. 1 I suggest to write TB_p,2 [note

that I use "_" to denote a sub-script] for the brightness temperature of polarization p=h or v and layer 2; you could keep the super-script d and u for down- and upwelling components. But in order to separate it more clearly from the d_1 I suggest to either use up- or downward pointing arrows or write "down" and "up" instead of "d" and "u". I don't see a reason why the physical temperature needs to have a sub-script 0. My suggestion: simply use T_2 with 2 being the sub-script denoting layer 2. I also suggest to use the sub-scripts denoting the boundaries between the layers not as sub-scripts to the index denoting the polarization. Hence, I suggest to write Gamma_p,12 and Gamma_p,23 instead of Gamma_p_12 for instance.

Line 136: "layers" –> "layer"

Line 194: "Figs. Fig." –> "Figs"

Lines 306 / 316: "east" –> "west"

Line 329: "Fig. 18": Could this be Fig. 14?

––––––––––––––––––––––––––––